# Effect of Temperature and Storage on Coffee’s Volatile Compound Profile and Sensory Characteristics

**DOI:** 10.3390/foods13243995

**Published:** 2024-12-11

**Authors:** Magdalena Gantner, Eliza Kostyra, Elżbieta Górska-Horczyczak, Anna Piotrowska

**Affiliations:** 1Department of Functional and Organic Food, Institute of Human Nutrition Sciences, Warsaw University of Life Sciences, 02-776 Warsaw, Poland; eliza_kostyra@sggw.edu.pl (E.K.); anna_piotrowska@sggw.edu.pl (A.P.); 2Department of Technique and Food Development, Institute of Human Nutrition Sciences, Warsaw University of Life Sciences, 02-776 Warsaw, Poland; elzbieta_gorska_horczyczak@sggw.edu.pl

**Keywords:** roasted coffee, storage, volatile compounds, sensory profile, espresso, cold brew

## Abstract

The study investigated the effects of storage temperature, type of coffee, and brewing method on coffee’s volatile compound profile and sensory quality. Three types of coffee were included in the study: Arabica, Robusta, and their 80/20 blend. Samples were stored at 5 °C and 20 °C for one month, after which the changes in the composition of volatile compounds were analysed and the sensory quality of espresso and cold brew coffee was assessed. The results showed that storing coffee at a lower temperature slows the changes in the profile of volatile compounds such as aldehydes, alcohols, pyrazines, and furans, helping preserve the desired aroma and flavour characteristics. Storage at higher temperatures resulted in greater changes in the volatile profile and sensory quality, with higher perceptions of earthy, sharp, and smoky notes and lower chocolatey and sweet notes. The brewing method also had a significant effect on the sensory quality. The espresso coffee had a higher intensity of coffee aroma, chocolate flavour, smoky aroma, and roasted notes. In contrast, cold brew coffee was perceived as sweeter, fruitier, and had more pronounced rum notes. The coffee type also significantly influenced the aroma and flavour profile. Arabica had a more harmonious and mild aromatic profile, while Robusta had a sharper aroma. The blend of Arabica and Robusta combined the characteristics of both coffees and offered a balanced aromatic profile.

## 1. Introduction

Of the more than 100 species in the genus *Coffea*, only two are economically important: *Coffea arabica* L., commonly known as Arabica, and *Coffea canephora* L., or Robusta, native to Congo [1]. The two species differ not only in the length of time required for ripening, their aroma, and colour but also in the composition and quantity of volatile compounds produced when the coffee beans are roasted. The aroma of *C. arabica* is characterised by a mild, non-astringent, and harmonious profile, whereas that of *C. canephora* is usually earthy and harsh. Robusta is typically utilised in the production of instant coffee [2]. The primary differentiating factor between Arabica and Robusta is the caffeine content, which is approximately twice as high in Arabica. Additionally, it is notable that chlorogenic acids are present in higher concentrations in Robusta, but a higher quantity of trigonelline—an alkaloid that enhances cognitive abilities, including memory—is observed in Arabica [3].

Coffee is appreciated not only for its caffeine content but also for its complex and inviting aroma, which is a symphony of volatile compounds released during the roasting process [4,5,6,7]. Volatile compounds are formed by several complex mechanisms resulting from a number of chemical reactions, such as Maillard and Strecker reactions, as well as the degradation of fats, sulphur amino acids, phenolic compounds, and many others [8]. In roasted coffee, about 30 compounds are responsible for the characteristic aroma [8]. The roasting process causes extreme changes in the beans’ chemical composition and affects their physical and sensory properties. Coffee beans become brown, dry, brittle, and porous [9].

From a food safety perspective, roasted coffee is sustainable when stored at less than 5% moisture. However, freshly roasted coffee is unstable and its composition changes after roasting. The chemical and physical changes underlying this deterioration are mainly CO_2_ loss and aroma degradation [10,11]. This gas release, also known as outgassing, occurs for about a month in whole-bean coffee, so appropriate packaging is used to prevent this [12,13]. Quantifying the loss of freshness of coffee is possible either by measuring the outgassing process or by analysing the change in coffee aroma from its original composition [11,13,14,15]. Advances in analytical methods, such as Headspace-GC/MS, have identified over 1000 volatile compounds in coffee beans. The complexity of coffee aroma is characterised by, among others, furans, pyrazines, aldehydes, formic and acetic acids, phenolic compounds or pyrroles, and thiophenes present at low concentrations [16,17,18]. By analysing aroma by GC/MS, the freshness of coffee can be assessed by monitoring the loss of aroma compounds or the appearance of new compounds [19,20]. Holscher and Steinhart (1992) [21] indicated that methanethiol has a strong influence on aroma freshness, with levels decreasing significantly as early as one day after roasting, while Czerny and Schieberle (2001) [22] reported that coffee staling is a result of 2-furfurylthiol degradation. However, the above methods for qualitative and/or quantitative VOCs (volatile organic compounds) composition are often time-consuming, labour-intensive, and expensive. An alternative is electronic nose, which is chemical-free, non-invasive, provides results in less time, and is accurate. E-nose has been widely used for identifying, typifying, and classifying food products [18,23,24,25]. It is essential in the case of coffee, as it allows the differentiation of original coffees with high-quality beans from cheaper varieties grown under inferior conditions but often fraudulently offered as fine coffees.

Despite the ubiquity of coffee consumption, many additional factors can influence its final aroma and sensory profile, such as storage time and conditions and brewing method. Understanding these factors ensures excellent coffee quality and satisfies increasingly demanding consumer tastes. For example, to prevent loss of aroma, some consumers have adopted the practice of storing coffee beans in the freezer, which should slow down any staling reactions or loss of volatiles, thus preserving the aroma of the beans compared to storage at room temperature. This practice was confirmed in studies by Mayer et al. (2000) [26] and Ismail et al. (2013) [27], which showed that low-temperature storage of brewed coffee and bags of ground coffee preserved the concentration of certain volatile compounds compared to storage at room temperature. Only a few studies have used roasted whole-bean coffee as a model [16,28,29,30]. In addition, the extraction method is also important in determining the final aroma of the coffee infusion, in turn significantly affecting the aroma profile [31], which we analyse using sensory methods. Attributes such as aroma, flavour, acidity, and body are sensory characteristics that describe the quality of coffee [32].

Studies on freshness loss and coffee ageing have mainly been conducted during the original coffee storage period [9,33,34,35]. In contrast, our research aimed to answer how the conditions of the secondary storage period after opening the coffee bean packages affect the qualitative changes in volatile organic compounds. The research also included determining the sensory profile of stored coffees brewed using espresso and cold brew. The variable factors were coffee type, storage, and brewing methods.

## 2. Materials and Methods

### 2.1. Materials

The research material consisted of two types of roasted coffee beans (100% Arabica, 100% Robusta, and a blend of 80% Arabica and 20% Robusta) sourced from Julius Meinl Poland, a representative of an Austrian brand of coffee and tea manufacturer in Poland. Data on coffee varieties and origins and the roasting profile of the coffee have not been obtained from the coffee roaster.

All samples for testing were packaged in dark original packaging with a valve to allow degassing of the interior. The samples were stored at room temperature (20 ± 1 °C) until the start of the tests. The samples were opened one month after roasting, immediately before the start of the tests. After the first measurement, the remaining coffee was sealed in foil bags. Half of the beans were stored in the refrigerator for one month, and the other half was kept at room temperature. VOC measurements were taken twice after the coffee had been opened and after the coffee had been stored for one month at room temperature and refrigerated conditions (5 ± 1 °C). In the first phase of the research, the beans of three types of coffee stored under different conditions were tested using an electronic nose. The study’s second phase involved a sensory evaluation of the coffee infusions from the samples analysed. For the espresso method, whole roasted coffee beans ground in an espresso machine were used to prepare the infusion (see below). In contrast, for the cold brew method, coffee ground to a particle size of 1.6 mm (coarse ground coffee) was used.

### 2.2. Methods

#### 2.2.1. Volatile Compounds Analysis of Coffee Beans

In the first phase of the study, the VOCs of each coffee were analysed using an electronic aroma analyser, the Herakles II (Alpha MOS, Toulouse, France). The electronic nose system consists of an analytical part based on an ultrafast gas chromatograph equipped with two columns of different polarity (mid-polarity DB-5 and nonpolar DB-1701) and control software with a chemometric section and the AroChemBase volatile compounds library. For each measurement, approximately 3 g of coffee beans were weighed and sealed in glass vials with a silicone/Teflon septum. All samples were placed in a blank tray (empty/control vials). The test was repeated three times for each type of coffee. The initial temperature was 40 °C, the isotherm was 5 s, and the acquisition time was 93 s. The injection volume was 1000 µL, the flow rate was 125 mL/s, and the injection temperature was 200 °C. The calibration method was described in previous studies [23,24].

#### 2.2.2. Coffee Brewing Methods

The sensory tests were conducted over two days, split between the espresso and cold brew brewing methods, first freshly opened from the factory packs and then after one month’s storage. For the extraction of espresso coffee, a Delonghi ECAM45X.6Y-45X.8Y Eletta Explore automatic coffee machine (manufactured by the Italian company De’Longhi (Treviso, Italy), bought in Poland) was used, with a 1450 W motor, frequency of 50~60 Hz, and a pressure of 19 bar per cup, with a capacity of two cups of coffee. According to the machine’s technical standard, a portion of 20 g of ground coffee was prepared with a grinder setting of 1. According to the technical specifications, this was considered a strong extraction, with a capacity of 80 mL for two cups of 2 × 40 mL [36]. The obtained brews were placed in appropriately labelled thermoses to maintain a constant temperature and then served for evaluation in coded glass beakers covered with watch glasses and placed in polystyrene warmers.

Ground coffee with a particle size of 1.6 mm (coarse ground) was used to extract the cold brew coffee. To obtain a coffee/water ratio similar to that of an espresso machine (1:4), 300 g of coffee was poured into 1200 mL of chilled 23–25 °C filtered water. The poured coffees were capped in glass bottles, shaken five times to soak all the coffee, and then stored at +5 °C for 24 h. After maceration, the contents of the bottles were mixed, and the coffee drink was filtered through a Toddy-Toddy^®^ Home Cold Brew System filter (Toddy, LLC, Loveland, CO, USA) to remove coffee particles.

#### 2.2.3. Sensory Evaluation of Coffee Brews

Quantitative Descriptive Analysis (QDA) was used to evaluate the sensory characteristics of the coffee samples, following the guidelines of PN-EN ISO 13299:2016 [37]. In the preliminary meeting, sensory experts established a list of attributes for the profiling procedure. Thirty descriptors were chosen and defined. The intensity of the attributes was measured on a linear scale from 0 to 10 cm, anchored from “none” to “very intense” for odour and taste/flavour descriptors and from “low” to “high” for overall odour intensity body and overall sensory quality (Table 1).

##### Sample Presentation Procedure

Individual coffee samples (30 mL) were prepared immediately before evaluation. Coffee was poured from a thermos into glass beakers (150 mL) coded with random 3-digit numbers, then covered with lids, and placed in temperature-maintaining containers. The samples were presented using a sequential monadic test. The temperatures of samples (60 ± 2 °C for espresso coffee brews and 5 ± 2 °C for cold coffee brews) were checked at the time of serving to ensure that the specified temperature was achieved. Mineral water served as a neutraliser between the evaluated samples.

##### Panel and Testing Conditions

The sensory evaluation was conducted by a trained sensory panel, qualified according to PN-EN ISO 8586:2014–03 [38]. The assessors had theoretical and practical knowledge of sensory procedures and the assessment of food products using different methods, including QDA. Each panellist participated in two sessions to ensure a thorough and reliable assessment. The evaluations were performed in the Laboratory of Sensory Analysis of the Institute of Human Nutrition Sciences at Warsaw University of Life Sciences. The laboratory meets the general requirements of the ISO standard (ISO8589:2010) [39]. The ten individual testing booths were equipped with the ANALSENS computerised system for data acquisition. The results of the profile assessment were converted into numerical values (conventional units—c.u.) using sensory software version 7.5. Each sample was analysed in two independent replicates, so the results were an average of 18 evaluations.

### 2.3. Statistical Analysis

The changes in the volatile profile were presented on a graph showing the differences in distances (conventional organoleptic units) between individual samples. The original software Alpha Soft (v. 8.0) was used for data processing. The results were statistically analysed with XLSTATS version 2021 (software by Addinsoft, Paris, France). Instrumental and sensory data were tested using variance analysis (ANOVA) followed by Fisher’s LSD significant test (at a 5% probability level).

One-way ANOVA analysis was carried out to determine differences between the volatile compounds of the examined coffee beans. The data were expressed as mean ± SD of three independent experiments. The profiling results were tested by a two-way analysis of variance (ANOVA), considering products, assessors, and interactions as fixed variables (sensory characteristics of coffee brews). To determine the similarities and differences in the sensory profiling of the evaluated coffee samples (the effect of temperature storage, coffee type, and two extraction methods), Principal Components Analysis (PCA) was used.

## 3. Results

### 3.1. Volatile Profiles of Coffee Beans

Table 1 shows the volatile compounds found in the three types of coffee studied. In the volatile compound profile for Arabica, 41 compounds were recognised, while 34 were recognised for Robusta. In the coffee blend (80% Arabica and 20% Robusta), 44 volatile compounds were recognised, all occurring in Arabica and Robusta. Studies often show Arabica has a richer volatile compound profile than Robusta [40,41]. The groups with the largest number of compounds were aldehydes (11) and N-compounds (8). The remaining groups of compounds were less numerous and included a total of seven compounds: esters, lactones, and ketone (three, three, and one, respectively); three alcohols; three acids; three S compounds; three terpenes; two furans and phenols; and three miscellaneous compounds. All recognised volatile compounds were present in the coffee blend. In 100% Arabica, benzeneacetaldehyde, dihydro-2(3H)-furanone, and 3-methyl-2-butene-1-thiol were absent. The 100% Robusta coffee was poorer in 10 volatile compounds, i.e., p-anisaldehyde, (E)-cinnamic acid, benzyl alcohol, p-cresol, [E]-whiskey lactone, 3-ethylphenol, 5-methyl-5(H)-cyclopentapyrazin, cymen-8-ol, maltol, and coumarin were absent (Table 2).

The total relative share of acids in the profiles of volatile compounds was 12%, 18% and 17% for Arabica, Robusta and blends, respectively. According to Yeager et al. (2021) [42], Robusta is richer in organic acids than Arabica. Acids give coffee an aroma called acetic, pungent. The group that has a significant impact on the profile of volatile compounds and also on the aroma of coffee is aldehydes [17]. Recognised aldehydes include acetaldehyde (ethereal, fresh, fruity), propanal (pungent, fruity), butanal (chocolate, cocoa, green), 3-methylbutanal (almond, apple, cheese), 2-methylbutanal (almond, apple, burnt), pentanal (fruity, almond, berry), 2-methylpentanal (cheese, earthy, fruity), benzeneacetaldehyde (cocoa, floral, grassy), n-nonanal (citrus, fatty), p-anisaldehyde (anise, floral, hawthorn, minty, sweet), and (E)-cinnamaldehyde (apple, candy, cinnamon).

Five of the seven N-compounds belong to the pyrazine group. Depending on the type of coffee, the relative percentage of pyrazines in all samples was significantly different in all three types of coffee. In the group of N-compounds, there was also pyridine, which is found in coffee as a result of the pyrolysis of trigonelline during the roasting of coffee beans.

Our study found only two compounds from furans, i.e., 2-methylfuran and 2-furanmethanol. The miscellaneous components of the volatile profile of coffee were compounds with a small share. The largest share was maltol, which chemically belongs to alcohols. Maltol was found in Arabica and the blend. During storage at 20 °C, the share of maltol did not change significantly in Arabica but increased in the blend. At 5 °C, the share of maltol increased in Arabica, but in the blend, there was no significant change in this compound compared to its share in fresh coffee.

Figure 1 shows changes in the chromatographic profiles of coffee volatile compounds using the distances obtained from the PCA analysis. On day 0, the difference between 100% Arabica (Ar) and 100% Robusta (Ro) coffee was about 60 units. After storage, the differences decreased to about 45 units. On the other hand, the difference between the Ar profiles and the 80% Arabica + 20% Robusta (ArRo) blend was smaller (about 40 units) than the difference between 100% Robusta (Ro) and the ArRo blend (about 90 units), which results from the higher share of Arabica in the coffee blend. During storage, the differences in the volatile compound profiles for Ar and ArRo decreased to about 10 units for both temperatures. The Ro profile during storage approached that of the ArRo blend from day 0.

### 3.2. Sensory Characteristics of Espresso Coffee Brews: Storage Effect

The results of profiling the espresso coffee brews differing in coffee types and temperature of storage are presented in Table 3. The examined coffee samples differed significantly in the intensity of attributes such as coffee odour (*p* = 0.004) and flavour (*p* = 0.002), chocolate odour (*p* = 0.037), sour taste (*p* < 0.001), sweet taste (*p* = 0.01), rum flavour (*p* = 0.029), and body impression (*p* = 0.011).

A similar intensity of coffee aroma was found in the coffee without storage (100% Arabica, 100% Robusta, 80% Arabica + 20% Robusta). No major changes in the perceptibility of the coffee aroma were observed when stored at 7 °C. It was noted that the 100% Robusta espresso stored at 20 °C had the lowest coffee aroma intensity after 1 month of storage, and the sample did not differ significantly from the following blends: 100% Arabica and 80% Arabica + 20% Robusta stored at 20 °C and 100% Robusta stored at 5 °C. The intensity of the chocolate odour was identical on day 0 in the samples, and they did not differ from the 100% Arabica espresso stored at temperatures of 20 °C and 5 °C. There was no storage effect on the changes in the intensity of the other odour attributes, regardless of coffee type (100% Arabica, 100% Robusta, Arabica–Robusta blends 80%:20%) and storage temperature. The overall intensity of odour attributes remained at the same level in the coffee espresso samples.

The coffee flavour was most pronounced in the 100% Arabica, 80% Arabica + 20% Robusta (without storage), and 80% Arabica + 20% Robusta stored at temperatures of 5 °C. These samples differed substantially from Robusta espresso on day 0 and coffee brews stored at various temperatures. Sour taste was more noticeable in coffees stored at 5 °C and 20 °C. The sour and sweet taste intensity was least perceptible in samples that were not stored. Similarly, the stored samples represented a higher intensity of rum flavour than the 0-day samples. The body of the examined coffees remained at a similar level (except Robusta espresso stored at 20 °C, with the lowest score). The coffee samples had almost the same mean score for overall sensory quality.

The influence of the stored temperature on changes in the sensory characteristics of the espresso coffee brews is presented on the PCA plot (Figure 2). It was stated that the first two principal components, Factor 1 (F1) and Factor 2 (F2), accounted for 66.10% of the total variance. The variability of the samples was linked to the greatest extent to the first principal component (46.65%), while the second component was associated with a smaller extent of variability (19.75%). Differences in the intensity of coffee odour and flavour, chocolate odour, fruity odour, sweet taste, rum flavour, and body were loaded mainly along F1. In contrast, differences in spicy odour and flavour, nutty flavour, earthy odour and flavour, fruity flavour, ash odour and flavour, and overall intensity odour were loaded along F2. The overall sensory quality of the coffee samples was positively related to the coffee odour and flavour. Coffee samples on day 0 represented different sensory properties to brews stored at 5 °C and 20 °C. The samples stored at 5 °C, such as 100% Arabica and 80% Arabica + 20% Robusta, were located near attributes like sour taste, sweet taste, and fruity and rum flavour. Other samples (100% Robusta) stored at 20 °C and 5 °C were found near fruit odour, acid odour, and ash flavour.

### 3.3. Sensory Characteristics of Cold Coffee Brews: Storage Effect

The results of the descriptive analysis (profiling) of the nine cold coffee brews differing in coffee types and storage temperature are presented in Table 4 Statistically significant differences in the intensity of the descriptors between evaluated coffee cold samples were noted for smoky odour (*p* < 0.001), earthy odour (*p* < 0.001), acid odour (*p* < 0.001), sweet odour (*p* = 0.036), rum odour (*p* < 0.001), fruity odour (*p* = 0.012), sharp odour (*p* = 0.004), earthy flavour (*p* = 0.032), astringency sensation (*p* < 0.001), pungency impression (*p* = 0.005), and overall sensory quality (*p* = 0.036).

The coffee samples represented a similar intensity of coffee and chocolate odour. A significantly higher intensity of smoky and earthy odour was found in the 100% Arabica and 100% Robusta cold brews stored at 20 °C for one month compared to other samples. At the same time, the 100% Robusta sample was found to have significantly the highest level of perceptible acid odour of all examined cold coffee brews. The samples of cold coffee such as 100% Arabica, 100% Robusta, blends of Arabica and Robusta (80:20), and 100% Robusta stored for a month at 5 °C were characterised by a significantly greater intensity of sweet odour than 100% Robusta stored at 20 °C and 100% Arabica stored at 5 °C. The 100% Arabica coffee did not differ significantly from other brews in rum odour. It was found that coffee blends stored at 20 °C and 5 °C represented the same intensity of the fruity odour as the other coffee samples. The sharp odour was more noticeable in coffee cold samples stored at 20 °C (higher average scores) and least perceived in the blend variant stored at 5 °C. Overall odour intensity remained similar in the evaluated coffees. The variation factors in the sensory research did not affect the perceptibility of most attributes in taste and flavour in evaluated coffee cold brews except earthy flavour, astringency, and pungency sensations. The 100% Arabica coffee stored for one month at 20 °C did not differ significantly from other brews in earthy flavour. Lower values of pungency ratings were found in all samples stored at 5 °C than in coffee brews not stored and 100% Arabica and 100% Robusta stored at 20 °C. It was found that the 100% Arabica coffee cold brew received significantly higher overall sensory quality than 100% Robusta on day 0, 100% Robusta stored at 20 °C, and 100% Arabica stored at 5 °C.

The similarities and differences in the sensory properties of coffee cold brews are presented on the PCA plot (Figure 3). The first two principal components, Factor 1 (F1, 42.51%) and Factor 2 (F2, 24.33%), accounted for 66.83% of the total variance. Differences in the perceptibility of mostly odour attributes such as chocolate, smoky, earthy, acid, sharp, and overall intensity were loaded mainly along F1, whereas the descriptors—coffee odour, fruit odour, rum odour, coffee flavour, rum flavour, and fruity flavour—were loaded along F2. It was found that 0-day 100% Arabica cold brew and coffee samples stored at 5 °C such as 100% Arabica and 80% Arabica + 20% Robusta were situated close to overall sensory quality. Nearby were coffee samples—100% Robusta (day 0), 80% Arabica + 20% Robusta (day 0), and 100% Robusta stored at 5 °C—characterised by attributes such as sweet smell and taste, chocolate smell and taste, and burnt flavour. The sample of coffee cold brews stored at 20 °C exhibited a distinctly different sensory profile.

### 3.4. Effect of Brewing Methods on Sensory Properties of Non-Stored Coffees Samples

The influence of the brewing methods on the sensory properties of examined coffee samples is displayed on the PCA plot (Figure 4). Factor 1 (F1) and Factor 2 (F2) accounted for 86.50% of the total variance. Of the variability, 64.25% was attributed to the first principal component. The second principal component accounted for 22.25% of the variability. Differences in the intensity of attributes: coffee odour, smoky odour, burnt odour and flavour, sweet odour and taste, rum odour and flavour, and fruity odour and flavour were loaded mainly along F1. Descriptors such as sharp odour, earthy flavour, ash flavour, and overall sensory quality were mainly associated with F2. It was found that 100% Arabica espresso and 80% Arabica + 20% Robusta espresso were situated near coffee odour, smoky odour and flavour, body, chocolate odour and flavour, and overall sensory quality. The 100% Robusta espresso was associated with a noticeable earthy odour, burn odour and flavour, sharp odour, ash odour, and overall intensity odour. A separate cluster was formed by coffee cold brewed samples with the intensity of fruity odour/flavour, sweet odour/flavour, and rum odour/flavour.

## 4. Discussion

Roasted coffee beans are rich in volatile compounds that shape the aroma profile of coffee. Over 1000 different compounds have been detected in roasted coffee, but only about 5% of them affect the aroma of coffee [43,44,45,46,47]. In the study of the volatile compounds of three types of coffee, 44 compounds were recognised. Many factors affect the quantity of volatile compounds in coffee, including the origin of the coffee, the method of roasting it, and the research methodology, hence resulting the different numbers of detected volatile compounds, e.g., from 40 to 80 [48,49,50]. Aldehydes are one of the largest groups of compounds that influence the volatile compounds profile of coffee [17]. The most numerous groups in our study were aldehydes and N-compounds. The remaining groups of compounds were less numerous. Most volatile compounds were present in all types of coffee, which is consistent with the results of other researchers [40,41,42].

Three organic acids were recognised in all tested coffees. Acids in coffee are responsible for the pungent and acetic aroma [40]. In our study, the higher share of acids in Robusta was mainly pentanoic and acetic acid. We found that Robusta was richer in organic acids than Arabica tends to exhibit more acidity than Arabica, and that acidity is influenced by roasting method, which is in line with the study by Freitas et al. (2024) [51]. During storage, the acidification of coffees was significantly lower when stored at 5 °C. Our observation is supported by studies showing that storage at higher temperatures contributes to coffee acidification [33,52].

A significant difference was found in the relative peak area shares of the three alcohols observed in Arabica, Robusta and their mixture. Colzi et al. (2017) [53] found that methanol may be one of the volatile compounds distinguishing Arabica from Robusta. The share of methanol in Arabica was significantly higher than in Robusta, similar to the study by Colzi et al. (2017) [53]. During storage at 5 °C, the share of methanol significantly decreased, in contrast to storage at 20 °C, where the share of methanol significantly increased. Methyl alcohol is a very volatile compound compared to other alcohols. It probably partially evaporated from coffee beans at a temperature of 20 °C. Benzyl alcohol was present only in Arabica and in the blend of coffee. An aromatic, floral, fruity, and sweet aroma characterises benzyl alcohol. Cantergiani et al. (2001) [54] and Lee and Shibamoto (2002) [55] found benzyl alcohol in unroasted Arabica beans and suggested that this compound may be a manifestation of insufficiently roasted coffee. The proportion of Benzyl alcohol in coffee stored at 20 °C was significantly higher than in fresh coffee. However, in coffee stored at 5 °C, the proportion of Benzyl alcohol decreased and was significantly lower than in fresh coffee and stored at 20 °C. Increased concentrations of alcohols, such as methanol and benzyl alcohol, may result from microbial fermentation and degradation of carbohydrates. Storage at lower temperatures (5 °C) allows for better preservation of the desired aroma [16,27,29,30].

The most numerous group of volatile compounds in coffee was aldehydes. Maeztu et al. (2001) [56] observed a similarly numerous group of aldehydes, identifying 70 compounds in espresso coffee using the GC-MS method, including eight aldehydes. Acetaldehyde and propanal were particularly responsible for the fruity flavour in this group. Caporaso et al. (2014) [57] identified 24 volatile compounds in Neapolitan coffee brew, espresso American, and moka coffee brews, including seven aldehydes. In our study, the total share of all 11 aldehydes in the volatile compound profile was 14%, 22%, and 19% for Arabica, Robusta, and blended coffees, respectively. The changes in the increase in the share of aldehydes during storage were significantly lower at 5 °C than at 20 °C. This observation is consistent with the conclusions of other researchers who found that, at higher temperatures, there is a faster loss of lighter volatile compounds, which also include shorter aldehydes [16,29]. Increased concentrations of aldehydes, such as acetaldehyde and benzaldehyde, are associated with lipid oxidation and amino acid degradation during storage. Higher temperatures accelerate these reactions, leading to greater changes in the aromatic profile of coffee [16,27,29,30].

The most significant number of compounds identified as N-compounds belong to the pyrazine group. Pyrazines are aroma compounds responsible for the aroma of roasted products [58]. Pyrazines are formed during the heat treatment of food in Maillard reactions, and they can also be products of the metabolic processes of microorganisms [59]. Their concentration may increase during storage due to further chemical reactions that are accelerated by higher temperatures [16,27,29,30]. Pyridine reduces the sensory quality of coffee through unpleasant aroma [60]. Pyrazines are a large group of compounds that contribute significantly to the aroma of coffee and coffee beverages [61,62,63]. The compound with a significant relative percentage was 1-methylpyrrole, which was present in a higher proportion in Arabica than in Robusta. Similarly, in the study of Ludwig et al. (2018) [61], they found a higher presence of 1-methylpyrrole in Arabica coffee (peak area 13,884 ± 602) than in Robusta (peak area 9870 ± 675) using SH-GC-MS. This compound contributes to the antioxidant activity of coffee. During storage, in most cases, compounds from the N-compound group decreased their percentage at 5 °C. This phenomenon was revealed for pyrazine (bitter, corn, hazelnut), pyridine (amine, fishy), 2-methyl pyrazine (potato, baked), and trimethyl pyrazine (baked, baked potato, bread). However, higher values were observed for coffees stored at 20 °C. Only 5-methyl-5(H)-cyclopentapyrazine (roast) increased its presence at both storage temperatures.

Furans are another significant group for coffee aroma. Our study found two furan derivatives, i.e., 2-methylfuran and 2-furanmethanol. 2-methylfuran gives coffee sensory features described as chocolate and sweet [62]. 2-furanmethanol is recognised as a characteristic compound shaping the aroma of coffee perceived sensorially as burnt sugar, caramelised [62]. Like Dippong et al. (2022) [63], 2-furanmethanol was one of the most abundant compounds in Arabica coffee, while it was less abundant in Robusta. In the current study, 2-furanmethanol was the compound with the highest relative percentage share in the volatile compounds profile in all coffee types. Moreover, during storage, the share of 2-furanmethanol increased significantly. However, the increase was significantly lower during storage at 5 °C. Furan and furan derivatives, such as 2-furanmethanol, are degradation products of carbohydrates and lipids. Their concentration increases during storage, especially at higher temperatures, which affects the changes in the aromatic profile of the coffee [29,30,56]. The opposite phenomenon was observed for 2-methylfuran, i.e., at 5 °C, the increase in relative share was significantly more significant than at 20 °C.

The miscellaneous components of the volatile profile of coffee were compounds with a small share, and the largest share was maltol, which chemically belongs to alcohols. Maltol is formed during coffee roasting due to the breakdown of maltose and may also manifest as over-roasted coffee [64]. Ethyl maltol, maltol, and coumarin are bioactive components in coffee. In Arabica and blend beans, a small amount of coumarin was found, which did not change during storage at 5 °C but decreased significantly at 20 °C. Coumarin belongs to the so-called “plant phenolics” and is also found in coffee [65,66,67]. The content of volatile compounds is variable and dependent on many factors. Dippong et al. (2022) [63] showed that volatile compounds are released differently depending on the type of coffee and degree of roasting. The most abundant volatile compounds present in the samples were furan, 2-methylfuran, methyl formate, 2,3-pentanedione, methylpyrazine, acetic acid, furfural, 5-methylfurfural, and 2-furanmethanol. Polyphenol content was slightly higher in Robusta than Arabica varieties and in more intensely roasted beans than medium roasted beans. Our results are consistent with previous studies by Mayer et al. 2000 [26] and Ismail et al. (2013) [27], who showed that storage of coffee at low temperatures allows for better preservation of the concentration of some volatile compounds compared to storage at room temperature.

In our work, we observed that changes in the volatile chemical content were generally much smaller during storage at 5 °C than at 20 °C, which is consistent with the observations of other researchers [29,30,41]. Bröhan et al. (2009) showed that higher storage temperatures accelerate the loss of lighter volatile compounds, affecting coffee’s freshness [16]. Similarly, Cotter and Hopfer (2018) found that storing coffee at lower temperatures slows down the changes in the volatile compound profile, which helps to preserve the desired aroma and flavour characteristics [29].

A study by Lin et al. (2019) [68] showed that the activation energy for Brownian motion of coffee particles is relatively independent of particle size, and the viscosity of coffee suspensions correlates with the diffusion coefficient gradient for Brownian motion. A study by Gmoser et al. (2017) [69] shows that Brownian movements of coffee particles significantly affect coffee’s foam stability and rheological properties. Our study showed that storing coffee at 5 °C slows changes in the profile of volatile compounds such as aldehydes, alcohols, pyrazines, and furans. These compounds are key to the aroma and flavour of coffee, and their stability at lower temperatures contributes to maintaining the desired sensory characteristics. Storing coffee at a low temperature (5 °C) can positively affect flavour stability due to reduced chemical reaction rates, reduced Brownian motion and reduced oxygen access. Together, these factors contribute to the integrity of volatile aromatic compounds and the stability of coffee flavour.

The flavour of coffee is determined by many factors, including cultivar, growing location, coffee cherry maturity, roasting process, storing condition, and brewing method [46]. Many coffee aroma compounds are labile or highly volatile, so the coffee flavour deteriorates after roasting [4,70]. Therefore, proper packaging and storage temperature are essential to maintain coffee freshness and slow staling. Particular attention requires a second storage period because after opening the package, the protective atmosphere changes, accelerating the loss of coffee freshness [8]. In the present study, we demonstrate the effect of coffee bean storage temperature after opening the package on the quality of brews obtained by two extraction methods (espresso and cold brew). The results indicate that in the case of cold brew coffee, storing beans after opening the package at a temperature of 5 °C positively affects the sensory quality of the infusion. The intensity of most of the assessed sensory attributes remained at a level similar to the infusions prepared immediately after opening the package. In contrast, infusions prepared from beans stored at a temperature of 20 °C showed different sensory characteristics, with a higher perceptibility of earthy, sharp, smoky notes and a lower perceptibility of chocolate and sweet attributes. This may be because storing coffee beans at a lower temperature (5 °C) significantly slows down changes in the profile of volatile compounds such as aldehydes, alcohols, pyrazines, and furans, which helps to preserve the desired aroma and flavour characteristics. These observations are consistent with the results of other studies, which have shown that higher storage temperatures accelerate the loss of more volatile compounds, thus affecting the freshness of coffee [16,29].

However, espresso coffees obtained from beans stored for 1 month at different temperatures showed a similar sensory profile. Concerning the control sample, they had a lower intensity of coffee odour and flavour, chocolate flavour, and increased intensity of sour taste and rum flavour, but this did not significantly affect their overall sensory quality. Differences in sensory characteristics in coffee infusions obtained by different brewing methods indicate the need for further research. It would be advisable to determine the profile of volatile compounds in the beans and the brews, which would deepen knowledge of the impact of secondary storage. There is a lack of information in the literature on the influence of secondary storage conditions on the sensory quality of coffee infusions. Research has mainly focused on changes in the flavour volatile compounds profile of coffee beans stored under different conditions [8] or on changes in the colour of ground coffee [12].

We also observed that the coffee brewing method (espresso and cold brew) evoked significant changes in the sensory properties of the examined samples. This is in line with literature data that both different coffee bean grinding degrees and brewing and serving temperatures are important factors influencing the sensory quality of coffee beverages [71]. In our study, cold brew coffees were perceived as sweeter in odour and taste, more fruity in odour and flavour, and rum in odour and flavour compared to espresso coffees. This has been confirmed in research, according to which the flavour profile of cold brew coffee differed considerably from the beverage prepared by hot extraction as an effect of the longer extraction time [72]. Consequently, cold brew coffees were sweeter and smoother as the components with sweet flavour are soluble even in cold water, whereas oils and acids are non-soluble substances [73]. It is stated that low total phenolic content makes cold brew coffee slightly sweet and tasty. On the one hand, phenols and their degradation products are mainly responsible for coffee’s sour and astringent flavour [74,75,76]. In the study of Cordoba et al. (2021), hot coffees presented a higher level of bitterness, astringency, and acrid flavour than cold brew coffees [77]. In our study, fresh espresso coffee (100% Arabica and 80% Arabica + 20% Robusta) was more intense in astringency, coffee odour and flavour, smoky odour and flavour, and chocolate odour and flavour than cold brew coffee samples. Robusta espresso coffee also had a different profile, being more burnt in odour and flavour with a higher level of earthy odour than other assessed espresso samples and cold brew coffee variants. The differences in sensory profiles may be due to the different extraction methods, where the longer extraction time at a lower temperature for cold brew allows sweet-tasting ingredients to dissolve. At the same time, the oils and acids responsible for the bitter and astringent taste are less soluble in cold water [77]. In conclusion, our study confirms that the brewing method significantly impacts coffee’s sensory profile, which is in line with the results of Cordoba et al. (2021) [77].

## 5. Conclusions

The study showed that storing coffee at a lower temperature (5 °C) slowed changes in the profile of volatile compounds such as aldehydes, alcohols, pyrazines, and furans, helping to preserve the desired aroma and flavour characteristics. Storage at a higher temperature (20 °C) resulted in greater changes in the volatile profile and sensory quality, with increased perception of earthy, spicy, and smoky notes and reduced perception of chocolate and sweet notes. Coffee type also had a significant effect on the aroma and flavour profile. Arabica was characterised by higher levels of methanol and unique compounds such as p-anisaldehyde and (E)-cinnamaldehyde, giving it a more harmonious and milder aromatic profile. Conversely, Robusta had higher levels of organic acids, such as pentanoic and acetic acids, giving it a sharper aroma. The blend of Arabica and Robusta combined the characteristics of both coffees and offered a balanced aromatic profile. The extraction method also had a significant impact on the sensory quality of the coffee. Espresso was characterised by a higher intensity of coffee aroma, chocolate flavour, smoky aroma, and roasted notes. At the same time, cold brew was perceived as sweeter, fruitier, and had more pronounced rum notes. Cold brew was also softer and sweeter, with less astringency intensity and sharp notes. There were significant changes in the volatile compound profile after one month of coffee storage, regardless of temperature. The sensory characteristics of the samples depended on the coffee brewing method. Cold brew coffee brews prepared from coffee beans stored at 5 °C showed similar sensory properties to unstored samples. On the one hand, the sensory quality of espresso brews, regardless of storage temperature, was similar and different from fresh samples. An increase in the proportion of aldehydes and furans and a decrease in the proportion of alcohols and pyrazines altered the aroma and flavour of the coffee, which could lead to a deterioration in its sensory quality. In conclusion, both storage and temperature and the coffee type and extraction method significantly affect coffee’s volatile compound profile and sensory characteristics. Storing coffee at a lower temperature and choosing an appropriate extraction method can significantly improve its aroma and flavour quality.

## Figures and Tables

**Figure 1 foods-13-03995-f001:**
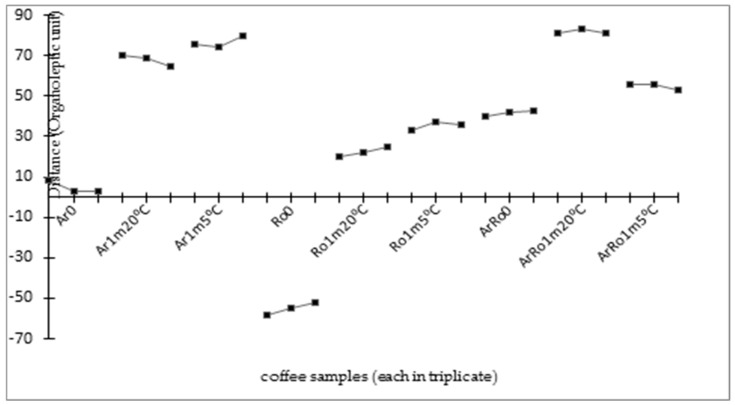
Differences in the chromatographic profiles of volatile compounds of three types of coffee (in triplicate) presented by PCA distances. Sample designations: Arabica (Ar), Robusta (Ro), and a blend of coffees with a composition of 80% Ar + 20% Ro (ArRo). All coffees were analysed on day 0 (0), after 1 month of storage at room temperature 20 ± 1 °C (1 m, 20 °C), and after 1 month of storage at low temperature 5 ± 1 °C (1 m, 5 °C).

**Figure 2 foods-13-03995-f002:**
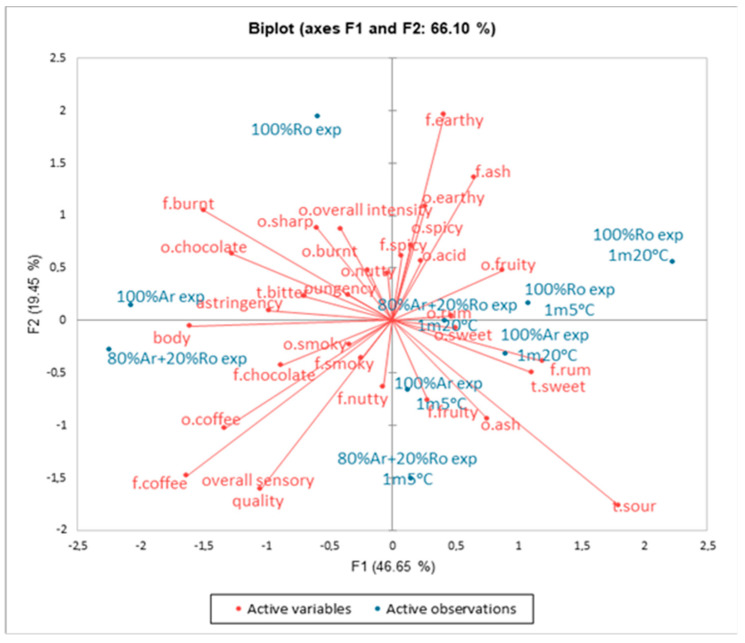
Similarities and differences in the sensory profile of coffee espresso brews on day 0 and after 1 month of storage at temperatures of 20 °C and 5 °C. Legend: 100% Arabica espresso day 0 (100% Ar exp), 100% Robusta espresso day 0 (100% Ro exp), 80% Arabica + 20% Robusta espresso day 0 (80% Ar + 20% Ro exp), 100% Arabica espresso 1 month 20 °C (100% Ar exp 1 m 20 °C), 100% Robusta espresso 1 month 20 °C (100% Ro exp 1 m 20 °C), 80% Arabica + 20% Robusta espresso 1 month 20 °C (80% Ar + 20% Ro exp 20 °C), 100% Arabica espresso 1 month 5 °C (100% Ar exp 1 m 5 °C), 100% Robusta espresso 1 month 5 °C (100% Ro exp 1 m 5 °C), 80% Arabica + 20% Robusta 1 month 5 °C (80% Ar + 20% Ro exp 5 °C).

**Figure 3 foods-13-03995-f003:**
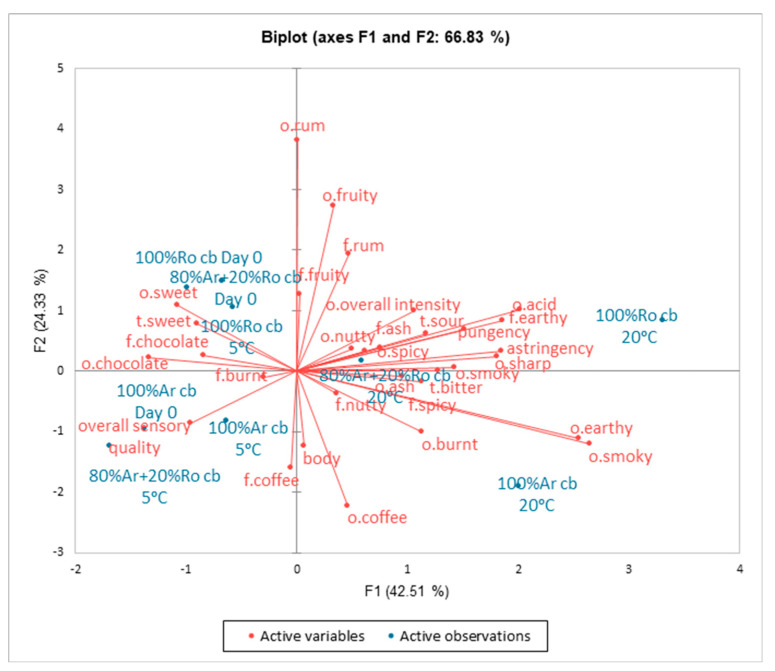
Similarities and differences in the sensory profile of coffee cold brews on day 0 and after 1 month of storage at temperatures 20 °C and 5 °C. Legend: 100% Arabica cold brew day 0 (100% Ar cb), 100% Robusta cold brew day 0 (100% Ro cb), 80% Arabica + 20% Robusta cold brew day 0 (80% Ar + 20% Ro cb), 100% Arabica cold brew 1 month 20 °C (100% Ar cb 1 m 20 °C), 100% Robusta cold brew 1 month 20 °C (100% Ro cb 1 m 20 °C), 80% Arabica + 20% Robusta cold brew 1 month 20 °C (80% Ar + 20% Ro cb 20 °C), 100% Arabica cold brew 1 month 5 °C (100% Ar br 1 m 5 °C), 100% Robusta cold brew 1 month 5 °C (100% Ro cb 1 m 5 °C), 80% Arabica + 20% Robusta cold brew 1 month 5 °C (80% Ar + 20% Ro cb 5 °C).

**Figure 4 foods-13-03995-f004:**
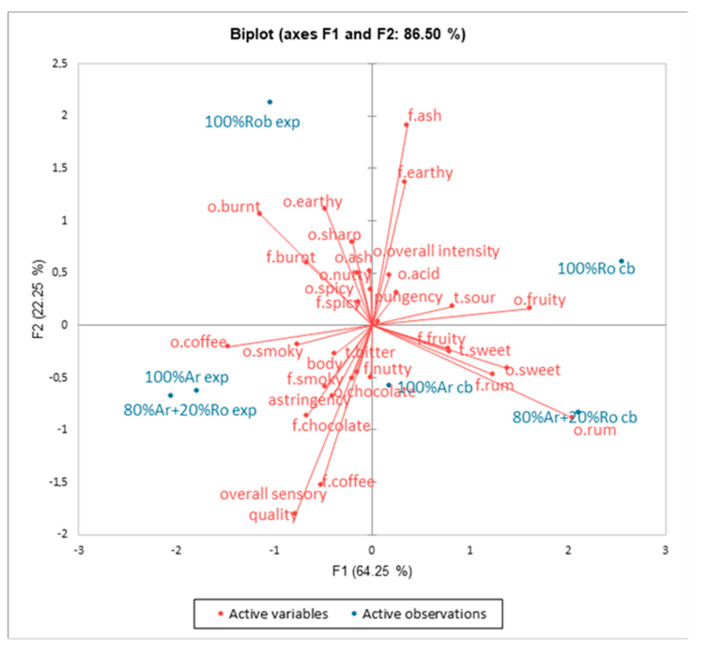
Similarities and differences in the sensory profile of coffee espresso and coffee cold brews on day 0. Legend: 100% Arabica espresso day 0 (100% Ar exp), 100% Robusta espresso day 0 (100% Ro exp), 80% Arabica + 20% Robusta espresso day 0 (80% Ar + 20% Ro exp), 100% Arabica cold brew day 0 (100% Ar cb), 100% Robusta cold brew day 0 (100% Ro cb), 80% Arabica + 20% Robusta cold brew day 0 (80% Ar + 20% Ro cb).

**Table 1 foods-13-03995-t001:** Descriptors and their definitions used in the sensory evaluation of coffee samples.

Attributes	Definitions
o.coffee	odour characteristic of roasted coffee brews
o.chocolate	odour associated with dark chocolate
o.smoky	odour related to the product of combustion of wood or leaves
o.ash	odour associated with the residue of burnt cigarette
o.burnt	odour related to the dark brown sensation of an over-roasted product
o.earthy	odour associated with long-stored potatoes in the cellar, slightly musty notes
o.acid	odour associated with vinegar
o.sweet	odour characteristic of honey, mild, and flowers
o.rum	odour related to dry fruity, oak wood, and caramel
o.fruity	odour associated with a blend of ripe fruits (e.g., peaches, pineapple)
o.nutty	odour characteristic of roasted nuts
o.spicy	odour associated with spices (e.g., cloves, cinnamon)
o.sharp	odour causing a sensation of irritation
o.overall intensity	overall orthonasal intensity of common coffee aroma
f.coffee	flavour characteristic of roasted coffee brews
f.chocolate	flavour associated with dark chocolate
f.smoky	flavour related to something scored or burnt
f.burnt	flavour related to the dark brown sensation of an over-roasted product
f.earthy	flavour associated with long-stored potatoes in the basement, slightly musty notes
f.ash	flavour of ash (e.g., an ashtray, cigarette smoke)
t.sour	basic taste of sour associated with a citric acid solution
t.bitter	basic taste of bitter associated with a caffeine solution
astringency	a mouth-drying sensation, puckering feeling
pungency	a burning sensation in the oral cavity
t.sweet	basic taste of sweet related to a sucrose solution
f.rum	flavour related to dry fruity, oak wood, and caramel
f.fruity	flavour characteristic of blended ripe fruits (e.g., peaches, pineapple)
f.nutty	flavour characteristic of roasted nuts
f.spicy	flavour characteristic of spices (e.g., cloves, cinnamon)
body	viscosity of the coffee, a sensation of fullness, and an opposite impression of diluted coffee
overall sensory quality	sensation of the harmony of the perceived attributes, with no or only slight intensity of negative notes

**Table 2 foods-13-03995-t002:** The relative percentage of peak areas in the chromatographic profiles of volatile compounds identified in fresh Arabica (100%), Robusta (100%), and blended coffees (Arabica 80%: Robusta 20%) and stored for one month at 20 °C and 5 °C.

Compound	IR DB-5	Relative Percentage of Peak Areas in the Chromatographic Profiles of Volatile Compounds (% ± SD)									*p*-Value
		Day 0			1 month at 20 ± 1 °C			1 month at 5 ± 1 °C			
		100% Arabica	100% Robusta	80% Arabica + 20% Robusta	100% Arabica	100% Robusta	80% Arabica + 20% Robusta	100% Arabica	100% Robusta	80% Arabica + 20% Robusta	
1	2	3	4	5	6	7	8	9	10	11	12
ACIDS											
Acetic acid	590	1.12 ± 0.1 ^cd^	1.62 ± 0.13 ^a^	1.23 ± 0.08 ^bc^	1.02 ± 0.02 ^de^	1.00 ± 0.02 ^e^	0.99 ± 0.02 ^e^	1.12 ± 0.02 ^cd^	1.51 ± 0.05 ^a^	1.26 ± 0.05 ^b^	<0.0001
Butanoic acid	819	4.78 ± 0.01 ^a^	2.22 ± 0.10 ^e^	4.49 ± 0.11 ^ab^	4.54 ± 0.34 ^a^	2.14 ± 0.08 ^ef^	4.19 ± 0.19 ^bc^	4.08 ± 0.38 ^c^	1.86 ± 0.12 ^f^	3.37 ± 0.07 ^d^	<0.0001
Pentanoic acid	933	6.26 ± 0.09 ^e^	14.25 ± 0.81 ^b^	12.02 ± 0.75 ^c^	14.6 ± 1.04 ^b^	17.96 ± 0.37 ^a^	14.67 ± 0.31 ^b^	9.64 ± 0.17 ^d^	11.75 ± 0.07 ^c^	9.80 ± 0.15 ^d^	<0.0001
ALDEHYDES											
Acetaldehyde	415	0.67 ± 0.02 ^e^	1.14 ± 0.18 ^d^	0.96 ± 0.04 ^d^	0.29 ± 0.01 ^f^	0.26 ± 0.01 ^f^	0.55 ± 0.03 ^e^	1.51 ± 0.18 ^c^	2.73 ± 0.21 ^a^	1.82 ± 0.06 ^b^	<0.0001
Propanal	480	5.65 ± 0.06 ^a^	4.99 ± 0.48 ^b^	5.20 ± 0.09 ^b^	2.28 ± 0.04 ^a^	1.45 ± 0.05 ^f^	2.33 ± 0.11 ^e^	3.98 ± 0.17 ^e^	4.55 ± 0.55 ^c^	3.70 ± 0.14 ^d^	<0.0001
Butanal	555	2.09 ± 0.08 ^b^	3.36 ± 0.09 ^a^	2.13 ± 0.10 ^b^	2.12 ± 0.123 ^b^	0.48 ± 0.02 ^f^	0.74 ± 0.03 ^e^	0.51 ± 0.03 ^f^	1.38 ± 0.15 ^c^	1.18 ± 0.04 ^d^	<0.0001
3-methylbutanal	655	0.09 ± 0.01 ^c^	0.16 ± 0.03 ^a^	0.12 ± 0.01 ^b^	0.09 ± 0.0 °C	0.03 ± 0.1 ^e^	0.09 ± 0.02 ^c^	0.09 ± 0.01 ^c^	0.04 ± 0.01 ^de^	0.06 ± 0.03 ^d^	<0.0001
2-methylbutanal	662	1.71 ± 0.02 ^bc^	2.44 ± 0.14 ^a^	1.83 ± 0.06 ^b^	0.71 ± 0.03 ^g^	1.00 ± 0.06 ^f^	0.97 ± 0.1 ^f^	1.18 ± 0.04 ^e^	1.60 ± 0.08 ^c^	1.37 ± 0.07 ^d^	<0.0001
Pentanal	701	2.57 ± 0.04 ^b^	1.90 ± 0.11 ^d^	3.67 ± 0.12 ^a^	1.94 ± 0.01 ^d^	0.99 ± 0.08 ^f^	1.88 ± 0.20 ^d^	2.23 ± 0.06 ^c^	1.37 ± 0.06 ^e^	1.92 ± 0.14 ^d^	<0.0001
2-methylpentanal	747	1.03 ± 0.03 ^b^	1.33 ± 0.09 ^a^	1.02 ± 0.03 ^b^	0.94 ± 0.02 ^c^	1.30 ± 0.02 ^a^	0.99 ± 0.06 ^bc^	0.92 ± 0.05 ^c^	1.31 ± 0.02 ^a^	0.92 ± 0.04 ^c^	<0.0001
Benzeneacetaldehyde	1022	0.00 ± 0.00 ^f^	5.81 ± 0.17 ^b^	3.46 ± 0.11 ^d^	0.00 ± 0.00 ^f^	8.04 ± 0.06 ^a^	3.96 ± 0.04 ^c^	0.00 ± 0.00 ^f^	4.08 ± 0.24 ^c^	2.56 ± 0.07 ^e^	<0.0001
n-nonanal	1118	0.58 ± 0.02 ^d^	1.91 ± 0.08 ^b^	0.66 ± 0.04 ^cd^	0.72 ± 0.02 ^cd^	2.43 ± 0.23 ^a^	0.84 ± 0.06 ^c^	0.70 ± 0.02 ^cd^	1.83 ± 0.25 ^b^	0.77 ± 0.01 ^cd^	<0.0001
p-anisaldehyde	1233	0.31 ± 0.02 ^c^	0.00 ± 0.00 ^d^	0.30 ± 0.03 ^c^	0.41 ± 0.02 ^ab^	0.00 ± 0.00 ^d^	0.38 ± 0.02 ^b^	0.46 ± 0.04 ^a^	0.00 ± 0.00 ^d^	0.44 ± 0.07 ^a^	<0.0001
(E)-cinnamaldehyde	1258	0.35 ± 0.03 ^d^	0.00 ± 0.00 ^e^	0.67 ± 0.01 ^c^	0.41 ± 0.05 ^d^	0.00 ± 0.00 ^e^	0.85 ± 0.05 ^b^	0.38 ± 0.05 ^d^	0.00 ± 0.00 ^e^	0.95 ± 0.12 ^a^	<0.0001
ALCOHOLS											
Methanol	360	0.20 ± 0.01 ^d^	0.07 ± 0.01 ^f^	0.25 ± 0.01 ^c^	0.13 ± 0.00 ^e^	0.05 ± 0.00 ^f^	0.25 ± 0.01 ^c^	0.35 ± 0.03 ^b^	0.32 ± 0.01 ^b^	0.47 ± 0.02 ^a^	<0.0001
Benzyl alcohol	1031	0.87 ± 0.05 ^b^	0.00 ± 0.00 ^g^	0.66 ± 0.02 ^d^	1.03 ± 0.05 ^a^	0.00 ± 0.00 ^g^	0.80 ± 0.02 ^c^	0.57 ± 0.06 ^e^	0.00 ± 0.00 ^g^	0.49 ± 0.01 ^f^	<0.0001
p-cresol	1050	0.71 ± 0.01 ^c^	0.00 ± 0.00 ^f^	0.49 ± 0.02 ^e^	0.89 ± 0.06 ^a^	0.00 ± 0.00 ^f^	0.63 ± 0.04 ^d^	0.78 ± 0.01 ^b^	0.00 ± 0.00 ^f^	0.65 ± 0.02 ^d^	<0.0001
FURAN DERIVATIVES											
2-methylfuran	613	6.55 ± 0.30 ^d^	6.15 ± 0.27 ^d^	4.44 ± 0.08 ^e^	1.55 ± 0.11 ^fg^	1.21 ± 0.08 ^g^	1.87 ± 0.16 ^f^	10.86 ± 0.24 ^b^	8.99 ± 0.44 ^c^	14.36 ± 0.31 ^a^	<0.0001
2-furanmethanol	865	19.17 ± 0.22 ^e^	9.20 ± 0.18 ^f^	27.43 ± 1.56 ^cd^	27.77 ± 1.66 ^c^	19.56 ± 0.67 ^e^	32.84 ± 1.11 ^a^	25.80 ± 1.82 ^d^	20.24 ± 0.84 ^e^	30.08 ± 0.14 ^b^	<0.0001
ESTERS/LACTONES/KETONES											
Methyl propanoate	635	0.29 ± 0.01 ^cd^	0.23 ± 0.01 ^e^	0.20 ± 0.01 ^e^	0.26 ± 0.02 ^d^	0.22 ± 0.01 ^e^	0.22 ± 0.02 ^e^	0.31 ± 0.02 ^bc^	0.35 ± 0.03 ^a^	0.34 ± 0.02 ^ab^	<0.0001
Cyclopentanone	788	1.34 ± 0.06 ^d^	0.00 ± 0.00 ^f^	1.08 ± 0.06 ^e^	1.65 ± 0.09 ^b^	0.00 ± 0.00 ^f^	1.28 ± 0.08 ^d^	1.85 ± 0.09 ^a^	0.00 ± 0.00 ^f^	1.52 ± 0.04 ^c^	<0.0001
Dihydro-2(3 H)-furanone	909	0.00 ± 0.0 °C	0.93 ± 0.08 ^b^	0.89 ± 0.08 ^b^	0.00 ± 0.0 °C	1.03 ± 0.11 ^b^	0.97 ± 0.05 ^b^	0.00 ± 0.0 °C	1.36 ± 0.18	1.32 ± 0.14 ^a^	<0.0001
Ethyl 3-(methylthio)propanoate	1098	1.48 ± 0.03 ^e^	2.49 ± 0.11 ^b^	1.76 ± 0.06 ^d^	1.81 ± 0.06 ^d^	3.81 ± 0.06 ^a^	2.11 ± 0.07 ^c^	1.39 ± 0.05 ^e^	2.39 ± 0.08 ^b^	1.43 ± 0.03 ^e^	<0.0001
[E]-whiskey lactone	1308	0.48 ± 0.02 ^c^	0.00 ± 0.00 ^e^	0.32 ± 0.05 ^d^	0.59 ± 0.06 ^a^	0.00 ± 0.00 ^e^	0.44 ± 0.01 ^c^	0.54 ± 0.03 ^b^	0.00 ± 0.00 ^e^	0.48 ± 0.02 ^c^	<0.0001
Triacetin	1350	0.34 ± 0.04 ^d^	0.46 ± 0.04 ^c^	0.62 ± 0.04 ^b^	0.30 ± 0.04 ^d^	0.95 ± 0.08 ^a^	0.86 ± 0.06 ^a^	0.54 ± 0.03 ^bc^	0.85 ± 0.14 ^a^	0.89 ± 0.02 ^a^	<0.0001
γ-Nonalactone	1363	0.34 ± 0.02 ^a^	0.14 ± 0.01 ^c^	0.32 ± 0.01 ^ab^	0.35 ± 0.05 ^a^	0.15 ± 0.01 ^c^	0.31 ± 0.03 ^ab^	0.32 ± 0.02 ^ab^	0.15 ± 0.01 ^c^	0.29 ± 0.02 ^b^	<0.0001
PHENOLS											
Acetophenone	1048	0.73 ± 0.01 ^f^	0.92 ± 0.08 ^c^	0.82 ± 0.02 ^de^	0.80 ± 0.03 ^def^	1.36 ± 0.04 ^a^	0.87 ± 0.04 ^cd^	0.75 ^ef^ ± 0.02	1.12 ± 0.09 ^b^	0.83 ± 0.02 ^d^	<0.0001
3-ethylphenol	1176	0.76 ± 0.05 ^bc^	0.00 ± 0.00 ^d^	0.72 ± 0.03 ^c^	0.90 ± 0.15 ^ab^	0.00 ± 0.00 ^d^	0.88 ± 0.05 ^ab^	0.88 ± 0.13 ^ab^	0.00 ± 0.00 ^d^	0.74 ± 0.01 ^c^	<0.0001
N-COMPOUNDS											
Pyrazine	717	1.52 ± 0.01 ^a^	1.17 ± 0.09 ^b^	1.52 ± 0.07 ^a^	1.08 ± 0.11 ^bc^	0.90 ± 0.03 ^d^	1.39 ± 0.08 ^a^	0.99 ± 0.18 ^cd^	0.95 ± 0.02 ^cd^	1.44 ± 0.04 ^a^	<0.0001
1-Methylpyrrole	750	15.43 ± 0.27 ^a^	9.98 ± 0.26 ^d^	14.27 ± 0.34 ^b^	13.94 ± 0.58 ^b^	7.75 ± 0.32 ^f^	13.04 ± 0.35 ^c^	9.88 ± 0.72 ^de^	7.19 ± 0.25 ^f^	7.19 ± 0.25	<0.0001
Pyridine	754	0.57 ± 0.03 ^b^	0.62 ± 0.03 ^ab^	0.66 ± 0.01 ^a^	0.44 ± 0.06 ^c^	0.38 ± 0.02 ^d^	0.49 ± 0.05 ^c^	0.37 ± 0.03 ^d^	0.38 ± 0.02 ^d^	0.38 ± 0.02 ^d^	<0.0001
2-methylpyrazine	827	6.72 ± 0.02 ^e^	8.79 ± 0.20 ^b^	7.53 ± 0.19 ^d^	7.23 ± 0.30 ^d^	9.61 ± 0.31 ^a^	8.14 ± 0.34 ^c^	5.68 ± 0.44 ^f^	5.93 ± 0.07 ^f^	5.77 ± 0.09 ^f^	<0.0001
2,6-dimethylpyrazine	915	1.48 ± 0.02 ^bc^	1.43 ± 0.1 °C	1.41 ± 0.08 ^c^	1.81 ± 0.01 ^a^	1.83 ± 0.01 ^a^	1.82 ± 0.01 ^a^	1.48 ± 0.03 ^bc^	1.76 ± 0.10 ^a^	1.54 ± 0.07 ^b^	<0.0001
trimethylpyrazine	983	3.61 ± 0.14 ^bc^	3.37 ± 0.05 ^d^	3.46 ± 0.09 ^cd^	3.39 ± 0.13 ^d^	4.39 ± 0.17 ^a^	3.78 ± 0.06 ^b^	2.58 ± 0.14 ^f^	3.42 ± 0.1 °C ^d^	2.87 ± 0.10 ^e^	<0.0001
5-methyl-5(H)-cyclopentapyrazine	1126	1.28 ± 0.05 ^bc^	0.00 ± 0.00 ^d^	1.49 ± 0.11 ^ab^	1.56 ± 0.05 ^ab^	0.00 ± 0.00 ^d^	1.76 ± 0.11 ^a^	1.51 ± 0.05 ^ab^	0.00 ± 0.00 ^d^	1.15 ± 0.5 ^c^	<0.0001
S-COMPOUNDS											
Dimethyl sulfide	519	1.32 ± 0.01 ^a^	1.28 ± 0.06 ^a^	0.90 ± 0.03 ^c^	0.53 ± 0.03 ^a^	0.45 ± 0.03 ^f^	0.41 ± 0.01 ^f^	0.93 ± 0.13 ^e^	1.15 ± 0.03 ^b^	0.65 ± 0.04 ^d^	<0.0001
Thiophene	672	4.53 ± 0.09 ^b^	5.76 ± 0.58 ^a^	5.49 ± 0.22 ^a^	2.42 ± 0.07 ^e^	1.85 ± 0.13 ^f^	3.93 ± 0.13 ^c^	2.93 ± 0.08 ^d^	2.98 ± 0.03 ^d^	4.45 ± 0.12 ^b^	<0.0001
3-methyl-2-butene-1-thiol	848	0.00 ± 0.00 ^d^	1.89 ± 0.10 ^b^	0.38 ± 0.02 ^c^	0.00 ± 0.00 ^f^	2.00 ± 0.05 ^a^	0.38 ± 0.06 ^c^	0.00 ± 0.00 ^f^	1.88 ± 0.04 ^b^	0.34 ± 0.06 ^c^	<0.0001
TERPENES											
Alpha-Pinene	950	0.28 ± 0.02 ^bc^	0.31 ± 0.02 ^ab^	0.28 ± 0.02 ^bc^	0.32 ± 0.05 ^a^	0.33 ± 0.02 ^a^	0.33 ± 0.01 ^a^	0.22 ± 0.02 ^d^	0.27 ± 0.02 ^bc^	0.25 ± 00.01 ^cd^	<0.0001
sabinene	976	2.14 ± 0.02 ^d^	2.78 ± 0.09 ^c^	0.37 ± 0.01 ^fg^	2.69 ± 0.03 ^c^	3.33 ± 0.04 ^a^	0.42 ± 0.02 ^f^	2.01 ± 0.08 ^e^	2.89 ± 0.09 ^b^	0.28 ± 0.01 ^g^	<0.0001
Cymen-8-ol	1197	0.51 ± 0.03 ^c^	0.00 ± 0.00 ^e^	0.43 ± 0.01 ^d^	0.65 ± 0.04 ^a^	0.00 ± 0.00 ^e^	0.56 ± 0.01 ^b^	0.50 ± 0.02 ^c^	0.00 ± 0.00 ^e^	0.49 ± 0.02 ^c^	<0.0001
MISCELLANEOUS											
Maltol	1145	1.24 ± 0.04 ^d^	0.00 ± 0.00 ^e^	1.72 ± 0.10 ^b^	1.31 ± 0.03 ^d^	0.00 ± 0.00 ^e^	2.17 ± 0.16 ^a^	1.54 ± 0.03 ^c^	0.00 ± 0.00 ^e^	1.74 ± 0.07 ^b^	<0.0001
Ethyl maltol	1208	0.70 ± 0.03 ^d^	0.99 ± 0.14 ^c^	0.76 ± 0.05 ^d^	0.75 ± 0.07 ^d^	1.84 ± 0.05 ^a^	0.99 ± 0.06 ^c^	0.80 ± 0.05 ^d^	1.57 ± 0.08 ^b^	0.80 ± 0.03 ^d^	<0.0001
Coumarin	1503	0.14 ± 0.01 ^a^	0.00 ± 0.00 ^d^	0.12 ± 0.02 ^b^	0.08 ± 0.01 ^c^	0.00 ± 0.00 ^d^	0.08 ± 0.01 ^c^	0.14 ± 0.01 ^a^	0.00 ± 0.00 ^d^	0.12 ± 0.01 ^b^	<0.0001

a, b, c …—Mean values with different letters in rows are significantly different at *p* < 0.05 (Fisher’s LSD test); IR DB-5—retention indexes for DB-5 column.

**Table 3 foods-13-03995-t003:** Sensory characteristics of espresso coffee samples analysed on day 0, after 1 month of storage at room temperature 20 °C and after 1 month of storage at low temperature 5 °C.

Attributes	100% Arabica Espresso Day 0	100% Robusta Espresso Day 0	80% Arabica + 20% Robusta Espresso Day 0	100% Arabica Espresso 1 month at 20 °C	100% Robusta Espresso 1 month at 20 °C	80% Arabica + 20% Robusta Espresso 1 month at 20 °C	100% Arabica Espresso 1 month at 5 °C	100% Robusta Espresso 1 month at 5 °C	80% Arabica + 20% Robusta Espresso 1 month at 5 °C	*p*-Value
o.coffee	4.80 ^ab^	4.70 ^ab^	5.35 ^a^	4.32 ^bc^	3.81 ^c^	4.39 ^bc^	4.66 ^ab^	4.17 ^bc^	5.29 ^a^	0.004
o.chocolate	2.83 ^a^	2.43 ^abc^	2.56 ^ab^	2.46 ^ab^	1.51 ^c^	1.83 ^bc^	2.21 ^abc^	1.64 ^bc^	1.51 ^c^	0.037
o.smoky	2.04	2.03	2.64	1.94	1.98	2.34	2.78	2.48	1.98	0.509
o.ash	1.36	2.10	2.17	2.14	2.55	2.30	3.19	2.01	2.59	0.063
o.burnt	3.27	3.76	3.14	3.16	3.11	2.72	3.66	3.24	3.25	0.835
o.earthy	1.02	1.90	1.11	1.14	1.61	0.95	1.36	1.19	1.14	0.749
o.acid	1.13	1.47	1.23	1.54	1.37	1.27	0.92	1.36	1.13	0.945
o.sweet	0.62	0.81	0.69	1.19	0.99	1.13	0.91	0.98	0.84	0.788
o.rum	0.85	0.67	0.71	1.03	1.13	0.79	0.90	1.28	0.69	0.828
o.fruity	0.58	0.96	0.60	1.13	1.56	0.73	0.93	1.05	0.74	0.548
o.nutty	0.86	1.17	0.71	0.95	0.64	0.84	0.84	0.99	0.78	0.965
o.spicy	0.71	1.05	0.61	0.78	0.89	0.71	0.78	0.78	0.51	0.946
o.sharp	1.34	1.74	1.20	0.77	0.79	1.16	1.28	1.07	0.95	0.172
o.overall intensity	4.85	5.29	5.00	4.24	4.75	4.76	4.84	4.86	4.66	0.615
f.coffee	5.31 ^ab^	4.37 ^cde^	5.81 ^a^	4.56 ^bcde^	3.97 ^e^	4.77 ^bcde^	4.88 ^bcd^	4.24 ^de^	5.18 ^abc^	0.002
f.chocolate	2.63	1.96	2.61	2.78	1.71	2.03	2.45	1.52	1.79	0.129
f.smoky	2.72	2.09	2.53	2.53	2.14	2.51	2.33	2.43	2.15	0.890
f.burnt	4.56	4.74	4.56	3.55	3.33	3.94	4.04	3.66	3.68	0.221
f.earthy	1.00	3.19	1.32	1.14	1.83	1.40	1.41	1.63	0.72	0.239
f.ash	1.41	2.86	1.66	2.29	2.16	2.03	2.15	2.18	1.76	0.487
t.sour	1.93 ^d^	2.62 ^cd^	2.29 ^cd^	3.31 ^abc^	3.36 ^abc^	2.84 ^bcd^	3.75 ^ab^	3.85 ^ab^	4.07 ^a^	0.001
t.bitter	4.08	4.18	4.68	3.39	3.94	4.14	3.88	4.00	4.04	0.766
astringency	2.68	2.63	3.31	1.83	2.33	2.05	2.31	2.54	2.62	0.763
pungency	1.08	1.41	1.49	0.73	1.02	1.11	1.15	1.38	1.24	0.738
t.sweet	0.22 ^d^	0.26 ^cd^	0.29 ^cd^	0.96 ^ab^	1.31 ^a^	0.88 ^abc^	0.76 ^abcd^	0.68 ^bcd^	0.78 ^abcd^	0.010
f.rum	0.39 ^c^	0.44 ^bc^	0.33 ^c^	1.31 ^a^	1.23 ^a^	1.29 ^a^	0.98 ^abc^	1.18 ^ab^	0.76 ^abc^	0.029
f.fruity	0.43	0.58	0.77	0.86	0.88	0.42	1.24	0.64	1.03	0.332
f.nutty	0.88	0.64	0.87	1.03	0.64	0.82	1.15	0.83	0.94	0.940
f.spicy	0.76	0.99	0.86	0.93	0.88	0.94	0.69	1.91	0.50	0.960
body	5.98 ^a^	5.98 ^a^	6.0^4 a^	5.13 ^ab^	4.34 ^b^	5.12 ^ab^	5.86 ^a^	5.42 ^a^	5.61 ^a^	0.011
overall sensory quality	5.63	4.64	5.66	5.13	4.41	5.01	5.46	5.00	5.57	0.131

o.—odour; t.—taste; f.—flavour; a, b, c …—Mean values with different letters in rows are significantly different at *p* < 0.05 (Fisher’s LSD test).

**Table 4 foods-13-03995-t004:** Sensory characteristics of cold brew coffee samples analysed on day 0, after 1 month of storage at room temperature 20 °C, and after 1 month of storage at low temperature 5 °C.

Attributes	100% Arabica Cold Brew Day 0	100% Robusta Cold Brew Day 0	80% Arabica + 20% Robusta Cold Brew Day 0	100% Arabica Cold Brew 1 month at 20 °C	100% Robusta Cold Brew 1 month at 20 °C	80% Arabica + 20% Robusta Cold Brew 1 month at 20 °C	Arabica Cold Brew 1 month at 5 °C	Robusta Cold Brew 1 month at 5 °C	80% Arabica+ 20% Robusta Cold Brew 1 month at 5 °C	*p*-Value
o.coffee	4.71	3.59	3.67	4.56	4.31	4.03	4.16	3.41	4.64	0.163
o.chocolate	2.61	2.24	2.70	1.64	1.48	2.39	2.33	2.14	2.31	0.359
o.smoky	1.81 ^b^	1.53 ^b^	1.64 ^b^	3.35 ^a^	3.04 ^a^	1.89 ^b^	1.64 ^b^	1.56 ^b^	1.46 ^b^	<0.001
o.ash	1.88	1.96	1.69	2.36	2.60	1.35	2.23	2.04	1.51	0.506
o.burnt	2.81	2.26	2.25	3.20	3.13	2.18	2.48	2.34	2.44	0.419
o.earthy	1.09 ^b^	0.74 ^b^	0.88 ^b^	2.35 ^a^	2.49 ^a^	1.13 ^b^	1.21 ^b^	0.79 ^b^	0.74 ^b^	<0.001
o.acid	0.95 ^cd^	1.37 ^bcd^	1.64 ^bc^	1.63 ^bc^	2.65 ^a^	1.83 ^b^	1.18 ^bcd^	1.18 ^bcd^	0.82 ^d^	<0.001
o.sweet	1.84 ^a^	1.98 ^a^	1.95 ^a^	1.16 ^ab^	0.85 ^b^	1.48 ^ab^	1.01 ^b^	1.93 ^a^	1.54 ^ab^	0.036
o.rum	1.74 ^abc^	2.58 ^a^	2.78 ^a^	0.8 °C	2.13 ^ab^	1.45 ^bc^	1.02 ^c^	2.33 ^ab^	0.81 ^c^	<0.001
o.fruity	1.28 ^bc^	2.34 ^a^	2.00 ^ab^	1.03 ^c^	2.35 ^a^	1.80 ^abc^	0.98 ^c^	2.29 ^a^	1.61 ^abc^	0.012
o.nutty	1.14	0.79	0.61	0.78	1.43	0.90	0.63	1.22	0.91	0.223
o.spicy	0.94	0.65	0.83	0.65	1.46	0.86	0.51	0.83	0.98	0.159
o.sharp	1.21 ^bcd^	1.28 ^bcd^	1.06 ^cd^	1.83 ^ab^	2.10 ^a^	1.50 ^abc^	0.82 ^cd^	1.12 ^bcd^	0.71 ^d^	0.004
o.overall intensity	4.99	5.21	5.14	5.35	5.49	5.25	4.38	4.90	4.65	0.271
f.coffee	5.23	4.18	5.08	5.45	4.43	4.88	4.61	4.69	5.16	0.153
f.chocolate	1.88	1.53	2.29	1.34	1.56	1.61	2.11	1.94	1.90	0.517
f.smoky	2.72	2.13	2.11	2.98	2.79	3.14	1.69	2.69	1.86	0.067
f.burnt	4.16	4.08	3.86	3.36	3.98	3.66	4.09	3.21	3.99	0.756
f.earthy	1.25 ^c^	1.81 ^abc^	1.61 ^bc^	2.04 ^abc^	2.80 ^a^	2.63 ^ab^	1.41 ^c^	1.71 ^bc^	1.40 ^c^	0.032
f.ash	1.96	2.49	1.76	2.23	3.01	2.28	1.88	2.55	2.01	0.640
t.sour	2.31	2.72	3.36	3.36	3.58	3.60	2.98	2.89	2.20	0.184
t.bitter	4.21	3.99	4.51	4.76	4.88	4.68	3.69	4.23	4.27	0.788
astringency	2.85 ^ab^	2.29 ^bc^	2.74 ^ab^	2.46 ^abc^	3.11 ^a^	2.59 ^ab^	1.29 ^d^	1.22 ^d^	1.73 ^cd^	<0.001
pungency	1.38 ^a^	1.31 ^ab^	1.48 ^a^	1.34 ^a^	1.40 ^a^	0.93 ^abc^	0.66 ^c^	0.77 ^bc^	0.69 ^c^	0.005
t.sweet	0.79	1.05	0.93	0.61	0.46	0.49	0.96	1.57	0.81	0.064
f.rum	1.23	1.45	1.61	0.89	1.63	1.19	0.76	1.68	0.59	0.160
f.fruity	0.94	1.37	1.29	0.79	1.40	1.51	1.28	1.56	0.89	0.467
f.nutty	0.78	0.60	1.07	1.05	1.03	0.85	0.77	0.64	0.98	0.774
f.spicy	0.79	0.67	0.78	1.20	1.39	0.91	0.58	0.61	0.65	0.304
body	6.17	5.41	5.91	6.24	4.91	5.73	5.08	5.02	5.46	0.053
overall sensory quality	5.61 ^a^	4.69 ^bc^	5.24 ^ab^	5.00 ^ab^	4.13 ^c^	5.26 ^ab^	4.66 ^bc^	4.93 ^abc^	5.34 ^ab^	0.036

o.—odour; t.—taste; f.—flavour; a, b, c …—Mean values with different letters in rows are significantly different at *p* < 0.05 (Fisher’s LSD test).

## Data Availability

The original contributions presented in the study are included in the article, further inquiries can be directed to the corresponding author.

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
