# Peer review of "Effect of Temperature and Storage on Coffee’s Volatile Compound Profile and Sensory Characteristics"

_foods, 2024, doi:10.3390/foods13243995_

Round 1

Reviewer 1 Report

Comments and Suggestions for Authors

The authors describe an interesting investigation into the post-roasting changes of coffee during different storage conditions. This mostly verifies previous opinions that low temperature storage could be beneficial.

The following revisions could be considered:

- the roasting date of the samples is unclear

- the coffee variety is not stated

- line 28: Coffea should be in italics

- Lines 33-34: there should be qualifiers such as "usually" because there are good taste qualities of Canephora available on the market as well. The harsh taste is typically associated with deficits in processing

- Line 34: the price differences recently more or less disappeared

- Line 52: CO2, subscript the 2

- Lines 79, and throughout: delete brackets with year

- Lines 101-102: add varieties. Canephora from India is mostly not the variety robusta.

- Line 123 and possibly throughout: add space between number and units

- Line 136: do not spell out SI units such as g

- Line 155: is the unit of the sensory scale really centimeters?

- Lines 210-222: this appears repitions to the previous text

- Line 229 and throughout: use correct degree symbol

- Line 236: what is Magnifito

- Figs 2-4: check quality

Author Response

REVIEW 1

Dear Reviewer,

I would like to thank you sincerely for your insightful analysis and valuable comments on my manuscript. All suggested corrections have been taken into account, which has significantly improved the quality of the article.

I appreciate the time and effort put into the review. Your suggestions were extremely helpful and allowed for a better presentation of the research results.

Best regards,

Magdalena Gantner

Comments and responses

The authors describe an interesting investigation into the post-roasting changes of coffee during different storage conditions. This mostly verifies previous opinions that low temperature storage could be beneficial.

The following revisions could be considered:

- the roasting date of the samples is unclear 

Response. coffees for the study were opened one month after roasting

The samples were opened one month after roasting, immediately before the start of the tests. 

- the coffee variety is not stated

Response. Unfortunately, I do not have this knowledge, because I received the coffees directly from the coffee roaster as experimental coffees

- line 28: Coffea should be in italics

Response. It has been improved

- Lines 33-34: there should be qualifiers such as "usually" because there are good taste qualities of Canephora available on the market as well. The harsh taste is typically associated with deficits in processing 

Response. The aroma of C. arabica is characterized by a mild, non-astringent, and harmonious profile, whereas that of C. canephora is usually earthy and harsh.

- Line 34: the price differences recently more or less disappeared

Response. The aroma of C. arabica is characterized by a mild, non-astringent, and harmonious profile, whereas that of C. canephora is usually earthy and harsh. Robusta is typically utilized in the production of instant coffee.

- Line 52: CO2, subscript the 2

Response. The chemical and physical changes underlying this deterioration are mainly CO2 loss and aroma degradation 

- Lines 79 and all subsequent lines: please remove brackets with the year

Response. I do not understand why the brackets with the year of publication should be removed, as it is in accordance with the mdpi requirements.

- Lines 101-102: add varieties. Canephora from India is mostly not the variety robusta.

Response. We do not know the names of specific varieties and we got this information about the origin of the coffee directly from the coffee roaster. Therefore, I have amended the sentence.

The research material consisted of two types of roasted coffee beans (100 % Arabica, 100 % Robusta, and a blend of 80 % Arabica and 20 % Robusta) sourced from Julius Meinl Poland, a representative of an Austrian brand of coffee and tea manufacturer in Poland. Data on coffee varieties and origins and the roasting profile of the coffee have not been ob-tained from the coffee roasted.

- Line 123 and possibly throughout: add space between number and units

Response. For each measurement, approximately 3 g of coffee beans were weighed and sealed in glass vials with a silicone/teflon septum.

The test was repeated three times for each type of coffee. The initial temperature was 40 °C, the isotherm was 5 s, and the acquisition time was 93 s.

A portion of 20 g of ground coffee was prepared with a grinder setting of 1, according to the technical standard of the machine.

- Line 136: do not spell out SI units such as g

Response. It has been improved

- line 28: Coffea should be in italics

Response. It has been improved

- Line 155: is the unit of the sensory scale really centimeters?

Response. Yes, the intensity of attributes was measured on a 10 cm scale

- Lines 210-222: this appears repitions to the previous text

Response. Thank you for your attention. The sentence with aldehydes has also been reworded so as not to suggest repetition of information.

- Line 229 and throughout: use correct degree

Response. Throughout the article the degree symbol has been corrected: ᵒC has been replaced with °C.

- Line 236: what is Magnifito

Response. Magnifito is the name of a blend of Arabica and Robusta coffees, but it has been removed

Differences in the chromatographic profiles of volatile compounds of three types of coffee (in triplicate) presented by PCA distances. Sample designations: Arabica (Ar), Robusta (Ro), and a blend of coffees with a composition of 20% Ro + 80% Ar (ArRo). All coffees were analyzed on day 0 (0), after 1 month of storage at room temperature 20±1 ⁰C (1m20⁰C) and after 1 month of storage at low temperature 5±1 ⁰C (1m5⁰C).

- Figs 2-4: check quality

Response. The quality of the figures in the publication sent to the editor is very good, and their final appearance depends on the proofreader and editors.

Reviewer 2 Report

Comments and Suggestions for Authors

You will find a detailed response as a PDF file in the appendix

Comments on the Quality of English Language

The authors should check the spelling again. Words are sometimes BE sometimes AE (e.g. characterized-characterised. Most manuscript parts are written in AE, so use this to make it easy for changes.

The terms “expresso” and “espresso” are also used. Use only one spelling, I think authors should move away from expresso and just use espresso. I would agree with a single term, although I don't think “x” is appropriate.

Author Response

REVIEW 2

Dear Reviewer,

I would like to thank you sincerely for your insightful analysis and valuable comments on my manuscript. All suggested corrections have been taken into account, which has significantly improved the quality of the article.

I appreciate the time and effort put into the review. Your suggestions were extremely helpful and allowed for a better presentation of the research results.

Best regards,

Magdalena Gantner

Comments and responses

Reviewer comments:

Effect of temperature and storage on coffee's volatile com-1 pound profile and sensory characteristics

The manuscript (MS) provides a thorough overview of the influence of storage temperature and brewing process on volatile and sensory changes in different types of coffee during second-stage storage. The authors present a well-focused introduction and a clear method section. The results are basically structured well, but some rewording is required to clarify and strengthen the presentation of the findings. Unfortunately, the most crucial section—the discussion—lacks depth and a comprehensive analysis, with insufficient reference to existing literature. This is the weak point of the MS in its current form. Below is a detailed list of remarks that need to be addressed.

Figures + Tables:

Fig 1: The Magnifito blend is not mentioned elsewhere. It is unnecessary information; please maintain consistency in your sample naming.

Response. Changes in the volatile compounds profile of three types of coffee: Arabica (Ar), Robusta (Ro), and a blend of coffees with a composition of 20 % Ro + 80 % Ar (80Ar). All coffees were analyzed on day 0 (0), after 1 month of storage at room temperature 20±1ᵒC (1) and after 1 month of storage at low-temperature 5±1ᵒC

While I understand the space limitations, it is not ideal to introduce Table 1 and then Figure 1; Figure 1 should be placed before Table 1.

The figure is unclear. It is not appropriate to present this as a connected line graph, as there is no direct relationship between the previous results when three different sample types are shown. Additionally, what do the points in between represent? The 0/1/L1 setup results in 9 points, yet the figure does not accurately reflect this as there are more points of measurement. The results appear forced and lack coherence. As it stands, the figure does not add value.

Response.Figure 1 is secondary to Table 1, or more precisely, it is a picture of the chromatographic profile variation of individual coffees. The distances (from PCA analysis) show the differences between the chromatographic profiles. There are exactly 3x3x3=27 points on the graph. Three types of coffees were analyzed, in three different states (fresh, refrigerated, stored at room temperature) and for each coffee, tests were performed in triplicate. The continuous line connects the points (single profiles) illustrating the change in the volatile compound profile of individual samples. As per the reviewer's advice, we have removed lines between different samples.

Unfortunately, we did not include a description for Figure 1, which we are doing now.

Figure 1 shows changes in the chromatographic profiles of coffee volatile compounds using distances obtained from PCA analysis. On day 0, the difference between 100% Arabica (Ar) and 100% Robusta (Ro) coffee was about 60 units. After storage, the differences decreased to about 45 units. On the other hand, the difference between the Ar profiles and the 80% Arabica + 20% Robusta (ArRo) blend was smaller (about 40 units) than the difference between 100% Robusta (Ro) and the ArRo blend (about 90 units), which results from the higher share of Arabica in the coffee blend. During storage, the differences in the volatile compound profiles for Ar and ArRo decreased to about 10 units for both temperatures. The Ro profile during storage approached the ArRo blend from day 0.

Fig 2-4: The image resolution and quality are unacceptable. The image is blurry, and the frame has inconsistent thickness. The blue circles in the figure are not explained and are more confusing than self-explanatory.

Response. We apologize for the quality of the figures. This is not up to us. The figures in the manuscript we sent are of good quality but are less legible during text processing. We will try to improve and will ask the editors for better quality.

Tab 1: It is awkward to say "fresh" and then "and stored." Please rephrase the header.

Response. It has been improved

Table 1. The relative percentage of peak areas in the chromatographic profiles of volatile compounds identified in fresh Arabica (100 %), Robusta (100 %) and blended coffees (Arabica 80%: Robusta 20%) and stored for one month at 20 ᵒC and 5 ᵒC.

Additionally, clarify what the superscript letters represent, specify which post-hoc analysis was used, and provide the exact p-value. Why are "compound" and "IR DB-5" horizontal while the rest are vertical (this also applies to other tables)? The spacing in the first column causes single letters or word parts to shift to the next line—please check the spacing. - corrected

Response. a, b, c … – Mean values with different letters in rows significantly differ at p <0.05 (Fisher’s LSD test); IR DB-5 - retention indexes for DB-5 column.

Tab 2: t.sweet why p-Value only with 2 decimal spaces?

Response. Thank you for your remark. It has been corrected

Tab 3: also check p-value spaces. Some have 4 decimal spaces no consistency in the MS ?

Response. Thank you for your remark. It has been corrected.

Line specific remarks:

28-33 references are needed has been added has been added

Response.Torres, G. A. L., Campos, C. N., Salomon, M. V., Pantano, A. P., & Almeida, J. A. S. de. (2021). Coffea arabica L: History, phenology and climatic aptitude of the state of S\u00E3o Paulo, Brazil. Arq. Inst. Biol., 88, e00602020. https://doi.org/10.1590/1808-1657000602020

56 SPME is introduced, but GC/MS is the common technique, known since 1990. The abbreviation “SPME” is well-established, so either introduce GC/MS or clarify the context for SPME as well.

Response. It has been improved Advances in analytical methods, such as Headspace-GC/MS, have identified over 1,000 volatile compounds in coffee beans.

65 “and Schieberle” there is no year while there are years in brackets in all other cases

Response. It has been improved. While Czerny and Schieberle (2001) [21] reported that coffee staling is a result of 2-furfurylthiol degradation.

66 VOC might be a known abbr. but introduce this too has been improved

Response. However, the above methods for qualitative and/or quantitative VOCs (Volatile Organic Compounds) composition……

82 only a few studies – you are naming only one please refer to more references

Response. It has been added

Cotter, A. R., & Hopfer, H. (2018). The Effects of Storage Temperature on the Aroma of Whole Bean Arabica Coffee Evaluated by Coffee Consumers and HS-SPME-GC-MS. Beverages, 4(3), 68.

Pérez-Martínez, M., Sopelana, P., de Peña, M. P., & Cid, C. (2008). Changes in Volatile Compounds and Overall Aroma Profile during Storage of Coffee Brews at 4 and 25 °C. Journal of Agricultural and Food Chemistry, 56(8), 3145-3154.

Bröhan, M., Huybrighs, T., Wouters, C., & Van der Bruggen, B. (2009). Influence of Storage Conditions on Aroma Compounds in Coffee Pads Using Static Headspace GC–MS. Food Chemistry, 116(2), 480-483.

89 The phrase "on the other hand" is used incorrectly in all instances. To use this correctly, it should be preceded by "on the one hand." Please change this in lines 436, 445, 490, and 518.

Response. It has been improved

92 insert line break

Response. It has been improved

105 How long have the samples been stored? Since the authors aim to indicate an influence of storage on the aroma, this is essential information that should be provided before starting the trials. is written in the last sentence of the paragraph and has now also been added at the beginning of the study description

Response. The other half was kept at room temperature. VOC measurements were taken twice after the coffee had been opened and after the coffee had been stored for one month at room temperature and refrigerated conditions (+6-7°C).

Half of the beans were stored in the refrigerator for one month, and the other half was kept at room temperature.

111 (+6-7°C) – not acceptable please indicate a specific temp with an average accepted deviation as in line 105

Response. It has been improved. VOC measurements were taken twice after the coffee had been opened and after the coffee had been stored for one month at room temperature and refrigerated conditions (7 ± 1 °C).     

117 approximately 1.6 mm – is not acceptable as particle size is an important parameter for extraction and to redo the trials

Response. It has been improved. Ground coffee with a particle size of 1.6 mm (coarse ground) was used to extract the cold brew coffee.

119 “2.1 + 2.2” are both headed “materials” – change this please, 2.2 is methods

Response. It has been improved

120 “2.2.1.” why is Volatile Compounds Analysis capitalized? Nothing else is written in this style

Response. It has been improved

121 please highlight this instrument as e-nose as you say this is an important instrument used

Response.The electronic nose system consists of an analytical part based on an ultrafast gas chromatograph equipped with two columns of different polarity (mid-polarity DB-5 and nonpolar DB-1701) and control software with a chemometric section and the AroChemBase volatile compounds library.

132 – the samples have already been introduced there is no need for a full sample explanation again Response. It has been improved

134 no proper statement of instrument origin

Response. It has been improved. For the extraction of espresso coffee, a Delonghi ECAM45X.6Y - 45X.8Y Eletta Explore automatic coffee machine (manufactured by the Italian company De'Longhi, bought in Po-land) was used, with a 1450 W motor, frequency: 50 ∼ 60 Hz and a pressure of 19 bar per cup, with a capacity of two cups of coffee.

139 Square brackets are used for references, which this is not. Additionally, the webpage is a reference, so it should be included in the reference section.

Response. According to the machine's technical standard, a portion of 20 g of ground coffee was prepared with a grinder setting of 1. According to the technical specifications, this was considered a strong extraction, with a capacity of 80 ml for two cups of 2 x 40 ml (Delon-ghi Ecam, 2020).

143 approximately 1.6 mm – is not acceptable as particle size is an important parameter for extraction and to redo the trials

Response. Ground coffee with a particle size of 1.6 mm (coarse ground) was used to extract the cold brew coffee.

148 no proper statement of instrument origin

Response. After maceration, the contents of the bottles were mixed, and the coffee drink was filtered through a Toddy - -Toddy® Home Cold Brew System filter (Toddy, LLC, Loveland, CO, USA) was used to remove coffee particles.

153 It would be helpful to include a list of the descriptors. Consider adding this in the appendix to provide more information and add value for the reader.

Response. Thank you for your remark. We have added a table with attributes and their definitions in the supplementary material.

163 There is a general problem of writing °C in the MS. The font of the ° symbol differs from the previous instances; it appears like a superscript zero. Please check the font of °C in the whole MS

Response. It has been improved

196-197 Please paraphrase the sentence this is not understandable has been improved

Response. Using static headspace to study the volatile compounds in coffee, 44 compounds were identified. The most abundant groups were aldehydes (11 compounds) and N-compounds (8).

206-208 Unfortunately, that is incorrect. The table actually shows increases (or no consistent change at all) in the different alcohol levels. The authors should check there results again trying to highlight the most relevant and correct results.

Response. The paragraph has been deleted.

210-211 References for the flavor are needed. It is not acceptable to simply state that it is as is for aromas. Additionally, describing aldehydes with the term "aldehydic" is not appropriate, as readers unfamiliar with these compounds will not find this information valuable. The authors could refer to: “Flavor chemistry of beer: part ii: flavour and threshold of 239 aroma volatiles. MBAA TQ vol. 12, no. (3), 1975, pp. 151-168 Meilgaard, M.C.”

Response. The group that has a significant impact on the volatile profile and also on the aroma of coffee are the aldehydes [Huang et al., 2007]. The suggested citation has been used, thank you.

212 propanal is pungent and fruity

Response. It has been improved

218 largest number? – in line 210 aldehydes are described as the most numerous group – differ between number of analytes detected and concentration of analytes detected. -. – corrected

Response. It has been improved.  Five of the seven N-compounds belonged to the pyrazine group.

221: reference is needed

Response Zheng X.-Q., Nagai C., Ashihara H.: Pyridine nucleotide cycle and trigonelline (N-methylnicotinic acid) synthesis in developing leaves and fruits of Coffea arabica. Physiologia Plantarum 2004, 122: 404-411. doi.org/10.1111/j.1399-3054.2004.00422.x

225: “very high” is a judgement which is not allowed in the result presentation

Response. It has been improved. Their total relative share in the profiles of volatile compounds of coffee was high and amounted to: 25.7 %, 15.4 % and almost 32 % in Arabica, Robusta and blend, respectively.

229 °C another font again

Response. It has been improved.

246 All numbers in brackets should be highlighted that these represent the p-values. Numbers in brackets without an explanation of what they represent are not acceptable

Response. Thank you for your remark. It has been corrected.

249 don’t introduce samples again – also 258 (also inconsistent in naming for the blend)

Response. It has been improved. A similar intensity of coffee aroma was found in the coffee without storage (100% Arabica, 100% Robusta, 80% Arabica + 20% Robusta). No major changes in the perceptibility of the coffee aroma were observed when stored at 5 °C.

253 “cue” is a non-acceptable wording

Response. It has been improved.

However, it was noted that the 100 % Robusta expresso had the lowest coffee aroma intensity after 1 month of storage, and the sample did not differ significantly from the following blends: 100% Arabica and 80 % Arabica + 20 % Robusta stored at 20 °C and 100 % Robusta stored at 5 °C.

261 Stating the temperature for only one blend is not helpful, and the description is unclear, as it seems like a writing error. It should be revised to clearly specify the temperatures for each blend.

Response. It has been improved. The coffee flavor was most pronounced in the 100 % Arabica, 80 % Arabica + 20 % Robusta (without storage) and 80 % Arabica + 20 % Robusta stored at temperatures 5 °C

264 “sore” is the wrong word here

Response. Thank you for your remark. It has been corrected.

268 “nearly” is not scientific. Paraphrase this

Response. It has been improved.

The coffee samples had almost the same mean score for overall sensory quality.

279 “66.107%” why 3 decimal places?

Response. Thank you for your remark. It has been corrected.

286 – 289 Which attributes are these in detail? Although they are visible in the figure, please briefly list them here, as this makes it easier for the reader.

Response. According to the authors, the attributes are indicated in the manuscript

308 “earthy odour (0.0001),” why 4 decimal places? Same line 310

Response. Thank you for your remark. It has been corrected.

347 not far away is not scientific also this is a judgment while result presentation is allowed only
Response It has been improved

Nearby were coffee samples: 100% Robusta (day 0), 80% Arabica + 20% Robusta (day 0) and 100% Robusta stored at 5°C, characterised by attributes such as "sweet smell and taste, chocolate smell and taste and burnt flavour

390 What were the expectations regarding this? Does it align with the literature, and why was it this way?

Response Yes, it is consistent with the results of other researchers

392 What do these percentages refer to? They were not presented or introduced earlier, so why are they being discussed now? -

Response Three organic acids were recognized in all tested coffees. Acids in coffee are responsible for the pungent and acetic aroma [41]. In our study, the higher share of acids in Robusta was mainly pentanoic and acetic acid. Robusta was found to be richer in organic acids than Arabica [41]

395-396 Where do these results come from all of a sudden? Acids were only presented in the table, but not as a significant finding in the results section. This should have been addressed earlier if it is being discussed now.

Response A description of acids was introduced in the Results section:

The total relative share of acids in the profiles of volatile compounds was 12%, 18% and 17% for Arabica, Robusta and blends, respectively. According to Yeager et al. (2021) [41], Robusta is richer in organic acids than Arabica. Acids give coffee an aroma called acetic, pungent.

396-397 The statement is incorrect; the referenced authors did not demonstrate this, rather, you found results that can be explained by it. –

Response During storage, the acidification of coffees was significantly lower when stored at 5 ᵒC. Our observation is supported by studies that have shown that storage at higher temperatures contributes to coffee acidification [Cong et al., 2020] [Aung Moon et al., 2022]

402-404 And why is that? Methanol is an important factor, and the change should be explained. Additionally, the results presentation indicates something different, as it states that it decreases. Also tab 1 shows an increase of methanol

Response The mistake has been corrected.

Methyl alcohol is a very volatile compound compared to other alcohols. It probably partially evaporated from coffee beans at a temperature of 20 ᵒC

406-409 There is no adequate discussion on why this is the case; sources are needed. Why is this important? Sources are required to support this. Why is this new, and is it relevant for coffee?

Response Unfortunately, we found no studies on benzyl alcohol during coffee storage

414 again what are these numbers?

Response These numbers are the summed percentages of all 11 aldehydes in the volatile compound profile of coffees on day 0.

414-415 The question here would also be: Are these Maillard reaction products or fatty acid oxidation products, linear or branched aldehydes? This point should be discussed in more detail.

Response It has been improved

416-417 The speed cannot be substantiated based on the data; this MS is about concentrations, not the rate of decrease.

Response. It has been corrected

421: Now, that’s your part here to discuss in more chemical detail, coffee is fermented and roasted—where do the pyrazines come from? In coffee, they are likely formed through temperature in roasting. Which pyrazines would then be of interest? Sources are needed to support this.

Response. As the Reviewer pointed out, they are mainly formed through two pathways - synthesized by microorganisms and thermal treatment. We would like to point out that while it is possible to partially outline which pyrazines can be found in roasted coffee beans, it is impossible to choose which are the most important. We discuss their presence in the context of human sensory experience. The nature of coffee, which is a plant product, and its processing are factors that determine the final characteristics of the product. In addition, the evaluation of humans, as individuals, and their perception of a particular aroma molecule varies greatly. We believe that when dealing with the topic of food in a sensory context, we should take a slightly more flexible approach. We agree that evaluating, for example: the cascade of chemical reactions and their individual products can be valuable and necessary in certain studies. However, we want to emphasize that this article deals with the equally important and complex issue of combining the analysis of chemical compounds with the context of human sensation.

The complexity of this issue is described, for example, in an article, from a journal where the main research areas include analytical chemistry:

  • Le Quéré, J.-L.; Schoumacker, R. Dynamic Instrumental and Sensory Methods Used to Link Aroma Release and Aroma Perception: A Review. Molecules 2023, 28, 6308. https://doi.org/10.3390/molecules28176308.

423 positive or negative aroma influence of pyrazines?

Response We would like to thank the Reviewer for their comment. As we mentioned earlier, and in line with what can be found in the literature, we believe that a clear indication of whether pyrazines have a negative or positive effect on aroma is unfavorable in terms of sensory context. Additionally, we believe that this is only possible to a certain extent. Linear pyrazines (like hexanal) are responsible for more “grassy and fresh” aromas. Branched (like 12-methyltridecanal) - for aroma perceptions that are more “distinctive,” associated with strong-smelling products (even referred to as “beef-like”). However, this does not mean, in our opinion, that any of these compounds has, in the case of complex coffee aroma, an unambiguously non-negative or positive aroma. A similar approach to this topic and recognition of its complexity can be found, for example, in the article:

  • Mesurolle, J.; Saint-Eve, A.; Déléris, I.; Souchon, I. Impact of Fruit Piece Structure in Yogurts on the Dynamics of Aroma Release and Sensory Perception. Molecules 2013, 18, 6035-6056. https://doi.org/10.3390/molecules18056035.

This year, the journal Foods also published an article addressing the topic of flavor in (capsuled) coffee. We would like to note, the differences in the approach to describing the topic of flavor depending on the approach adapted to the scope of the journal (analytical chemistry/food chemistry):

  • Basile, G.; De Luca, L.; Calabrese, M.; Lambiase, G.; Pizzolongo, F.; Romano, R. The Lipidic and Volatile Components of Coffee Pods and Capsules Packaged in an Alternative Multilayer Film. Foods 2024, 13, 759. https://doi.org/10.3390/foods13050759.

While we understand the Reviewer's inquisitiveness, we believe that our approach benefits the subject and frames it well within the context of the scope of the journal.

425 The reference to the source does not make sense. Isn't this also a finding from the current study?

Response has been improved

427 why is that so? Discuss why it contributes to the antioxidative potential

Response It has been inproved

431 the sentence makes no sense at all

Response has been improved

However, higher values were observed for coffees stored at 20 ᵒC. Only 5-methyl-5(H)-cyclopentapyrazine (roast) increased its presence at both storage temperatures.

434 That’s a good and important finding, but why? Did others find more? Is this related to coffee quality? More discussion is needed

Response: We thank the reviewer for this question and his interest. To the best of our knowledge, the results obtained by other research teams vary widely. This is due to the nature of coffee as a raw material. The small changes in chemical composition that subsequently affect the aroma components are influenced by many factors: the conditions in which the plants were grown, how they were treated after harvesting, how they were processed and how they were brewed. Again, we emphasise that it is difficult to conclude that one volatile compound is more important than another for the "quality" of the aroma. Another question is whether this is reasonable. An aroma as complex as coffee should be evaluated as a composition - a panel of experts can do this. Of course, the opinions of individuals/consumers can be radically different. To a certain extent we can define aroma - its positive and negative aspects, but in the context of "quality" we believe that an approach with a certain margin of error and flexibility should be adopted. In our discussion, we have highlighted various aspects and cited literature to illustrate the point. 

445 – 446 and why did they increase? Again more discussion and literature referencing

Response: Has been improved in manuscript.

Furans, such as 2-furanmethanol, are degradation products of carbohydrates and lipids. Their concentration increases during storage, especially at higher temperatures, which affects the changes in the aromatic profile of the coffee.

In general, the article discusses the changes in the volatile compound profile of coffee during storage at different temperatures. In particular, the increases in the concentrations of some volatile compounds, such as aldehydes, alcohols, pyrazines and furans, are the result of chemical processes occurring during storage.

Causes of Increase

Aldehydes: The increase in aldehydes such as acetaldehyde and benzaldehyde is related to lipid oxidation and amino acid degradation during storage. Higher temperatures accelerate these reactions, leading to greater changes in the aromatic profile of coffee.

Alcohols: Increases in alcohols such as methanol and benzyl alcohol can result from microbial fermentation and carbohydrate degradation. Storage at lower temperatures (5°C) slows down these processes, allowing for better preservation of the desired aromatic characteristics.

Pyrazines: Pyrazines are compounds formed during the Maillard reaction during coffee roasting. Their concentration can increase during storage due to further chemical reactions that are accelerated by higher temperatures.

Furans: Furans, such as 2-furanmethanol, are degradation products of carbohydrates and lipids. Their concentration increases during storage, especially at higher temperatures, which changes coffee's aroma profile.

New citations in the article were inserted and included in the literature list.

451 references needed

Response. Has been improved and references have been added.

Pintać, D., Bekvalac, K., Mimica-Dukić, N., Rašeta, M., Anđelić, N., Lesjak, M., Orčić, D. (2022). Comparison study between popular brands of coffee, tea and red wine regarding polyphenols content and antioxidant activity. Food Chemistry Advances. 1, 100030. https://doi.org/10.1016/j.focha.2022.100030.

453 no discussion just a presentation of results

Response: Has been improved.

The content of volatile compounds is variable and dependent on many factors. Dippong et al. (2022) showed that volatile compounds are released differently depending on the type of coffee and degree of roasting. The most abundant volatile compounds present in the samples were furan, 2-methylfuran, methyl formate, 2,3-pentanedione, methylpyrazine, acetic acid, furfural, 5-methylfurfural, and 2-furanmethanol. Polyphenol content was slightly higher in Robusta than Arabica varieties and in more intensely roasted beans compared to medium roasted beans. Our results are consistent with previous studies by Mayer et al. (2000) and Ismail et al. (2013) who showed that storage of coffee at low temperatures allows for better preservation of the concentration of some volatile compounds compared to storage at room temperature.

455 you are presenting only one study in the MS

Response. It has been improved

456 not a discussion

Response. It has been improved

In our work, we observed that changes in the volatile chemical content were generally much smaller during storage at 5°C than at 20°C, which is consistent with the observations of other researchers [30,42, 60, 71]. Bröhan et al. (2009) showed that higher storage temperatures accelerate the loss of lighter volatile compounds, affecting coffee's freshness [16]. Similarly, Cotter and Hopfer (2018) found that storing coffee at lower temperatures slows down the changes in the volatile compound profile, which helps to preserve the desired aroma and flavor characteristics [60].

462 why is “particular” capitalized? has been corrected

Response. It has been corrected. There was a missing dot at the end of the previous sentence.

464 From now on, this is more of a conclusion than a "discussion." It should include more chemical background and relevant references to provide a deeper analysis

Response. It has been corrected

482 not correct, sweet taste is with p = 0.064 not significant

Response. The sentence concerns the influence of brewing methods on the changes in the sensory profiles of coffee brews.The differences in this respect can be observed on the PCA graph (Fig. 4).

484 what is the literature? There is no reference

Response. Thank you for your comment. The reference has been added

486 – 488 This is the first time there is an acceptable discussion. More of this level of analysis is needed throughout the entire manuscript.

Response. We tried to enrich the discussion of the results. We hope that it will meet the Reviewer's expectations. To gain a deeper understanding of the relationship between changes in the volatile compound profile and the sensory quality of coffee brews obtained by different methods, it would be beneficial to analyse the volatile compounds in coffee infusions. This is an interesting direction for further research as it was highlighted in the manuscript.

491 Why is the year of the reference in square brackets? The ones above are in round brackets.

Response. Thank you for your comment. It has been corrected

493 which type of espresso fresh or stored ones? If stored which temp. etc again more discussion please.

Response. Thank you for your comment. It has been corrected

495 – 497 and why? Again there is a need to discuss this and not only present it

Response. Thank you for your comment. It has been corrected

505-506 This is being mentioned or discussed for the first time here. It should have been included earlier.

Response: It has been improved

Thank you for drawing attention to this aspect. We agree that the issue under discussion should have been introduced earlier in the article to provide better context and consistency. In response to your suggestion, we have amended the text accordingly.

Introduction of a new topic earlier in the article: We have moved the introduction of the topic under discussion to the ‘Introduction’ and ‘Materials and methods’ sections so that the reader has a fuller picture of the context of the study from the outset.

516-517 So, is this proof that cold storage might have a positive influence on flavor stability? Factors such as lower chemical reactions (e.g., oxygen) or reduced Brownian molecular movement, etc., need to be discussed.

Response. Yes, the results of our study suggest that storing coffee at a low temperature (5°C) can positively affect flavour stability. Storage at a lower temperature slows down changes in the volatile chemical profile, which helps to preserve the desired aroma and flavour characteristics of the coffee. Thank you for your comment. It has been corrected.

Storing coffee at a low temperature (e.g. 5°C) slows down chemical reactions such as oxidation or reduced Brownian molecular movement. Lower molecular activity means that volatile aroma compounds are less likely to degrade, helping to maintain their integrity and affect flavour stability. This ensures that the coffee retains its desired aroma and flavour characteristics for a longer period of time. (Mayer et al.2000, Ismail et al. 2013, Perez-Martines et al. 2008, Cotter &Hopfer 2018, Brohan et al. 2009). A study by Lin et al. 2019 showed that the activation energy for Brownian motion of coffee particles is relatively independent of particle size, and the viscosity of coffee suspensions correlates with the diffusion coefficient gradient for Brownian motion. A study by Gmoser et al. 20017 shows that Brownian movements of coffee particles significantly affect the foam stability and rheological properties of coffee. Our study showed that storing coffee at 5°C slows changes in the profile of volatile compounds such as aldehydes, alcohols, pyrazines and furans. These compounds are key to the aroma and flavour of coffee, and their stability at lower temperatures contributes to maintaining the desired sensory characteristics. Storing coffee at a low temperature (5°C) can have a positive effect on flavour stability as a result of reduced chemical reaction rates, reduced Brownian motion and reduced oxygen access. Together, these factors contribute to the integrity of volatile aromatic compounds and the stability of coffee flavour.

Lin, C., Zhou, W., Hu, C., Yang, F., & Lee, S. (2019). Brownian motion and Einstein relation for migration of coffee particles in coffee suspensions. Journal of the Science of Food and Agriculture, 99(8), 3950-3956. DOI: 10.1002/jsfa.9620.

Gmoser, R., Bordes, R., Nilsson, G., Altskar, A., Stading, M., Loren, N., & Berta, M. (2017). Effect of dispersed particles on instant coffee foam stability and rheological properties. European Food Research and Technology, 243(1), 115-121. DOI: 10.1007/s00217-016-2723-4.

522-524 This is basic knowledge, as all parameters shown will influence the aroma profile. It reads more like an introduction than a concluding sentence

Response. We agree that the present conclusion may seem more like an introduction than a summary. To better highlight the findings of our study, we provide a more detailed conclusion , which highlights the key findings and their significance. Thank you for your comment. It has been corrected.

Reviewer 3 Report

Comments and Suggestions for Authors

The authors present an interesting work. However, some observations are included in the attached. Authors are invited to improve to obtain the publication category

Comments on the Quality of English Language

In some sections of the manuscript the language is not well used, it is hard to understant the ideas of the work.

Author Response

REVIEW 3

Dear Reviewer,

I would like to thank you sincerely for your insightful analysis and valuable comments on my manuscript. All suggested corrections have been taken into account, which has significantly improved the quality of the article.

I appreciate the time and effort put into the review. Your suggestions were extremely helpful and allowed for a better presentation of the research results.

Best regards,

Magdalena Gantner

Comments and responses

Line 10-11: Improve the sentence. I do not consider that the article analyses…. The authors are invited to change the scientific writing.

Response. The study looked at the effects of storage, temperature, type of coffee, and extraction method on coffee's volatile compound profile and sensory quality.

Line 12: Please use a space between the number and symbol: 5 °C and 20 °C; consider this observation for the entire document. Also, when the author uses %, adding a space between the number and symbol is important.

Response. It has been improved

Line 19: The author uses the term “Espresso” to Explain what you are referring to because, in the last lines, it was not observed this coffee material or the author referred to the method. If this is the case, update the lines where the methods were used

Response. It has been improved

Line 28-30-32-33-73-74: Add a reference to support the information Line 89-90: Improve the lines: The author's study, on the other hand,

Response. In contrast, our own research aimed to answer the question of how the conditions of the secondary storage period, after opening the coffee bean packages, affect the qualitative changes in volatile organic compounds analysed by the e-nose method.

Line 94-97: These lines could cause a conflict of interest; they will be deleted

Response. It has been improved

Line 101: Who or whom is Julius Meinl Poland? Explain better if it is a supermarket or other type of supplier.

Response. Julius Meinl is the representative of an Austrian brand of coffee and tea manufacturer in Poland, whose tradition dates back to 1862.

Line 109: Define VOC as a first time

Response. It has been improved. Volatile organic compounds.

Line 112-113: Improve the sentence; it is not clear.

Response. In the first phase of the research, the beans of three types of coffee stored under different conditions were tested using an electronic nose.

Results

In this section, the author presents the results of the study. The results are well-presented

Discussion

In concordance with the results, this section is well presented. However, the authors are invited to improve; please add information when you refer to your data being confirmed by some authors in the manuscript. In most cases, there is no more information about it; explain why your data will be consistent with the bibliography.

Response. Thank you for your remark. It has been corrected

Round 2

Reviewer 2 Report

Comments and Suggestions for Authors

My respect to the authors; by incorporating all the comments, the quality of the manuscript has significantly improved. I appreciate the detailed consideration of the feedback. From my perspective, there are a few minor spelling errors, which I have noted below. Finally, the authors should review the manuscript once more, as there are still instances where spaces need to be added before listing references.

line 34: "pro-file" one word

line 45 [°] makes no sense, check if there needs to be an reference or if this is a typo

line 115: delete approximately

148: "- -" delet one dash

154: Supplemental Table 1cneeds to be referenced somewhere in the manuscript, I think this is the appropriate spot

Table 1: "compound" and "IR DB-5" - Rotate this by 90° to align it with all tables and headings.

Alpha Pinene p-Value is unclear, 3 digits, is it lower 0.0001?

474: 2-furanmethanol is technically not a furan but an alcohol (maybe oxidized furan)

500: reference after "Cotter and Hopfer (2018)" not at the end of the sentence

Author Response

Dear Reviewer,

I would like to express my sincere gratitude for reviewing my article once again and for your valuable comments and suggestions. Your input has significantly contributed to improving the quality of the work. I am grateful for the time and effort you have dedicated to the review process.

Best regards,

Magdalena Gantner

Responses

line 34: "pro-file" one word

Response: has been corrected

line 45 [°] makes no sense, check if there needs to be an reference or if this is a typo

Response: has been corrected

line 115: delete approximately

Response: has been corrected

148: "- -" delet one dash

Response: has been corrected

154: Supplemental Table 1cneeds to be referenced somewhere in the manuscript, I think this is the appropriate spot

Response: has been corrected

Table 1: "compound" and "IR DB-5" - Rotate this by 90° to align it with all tables and headings.

Response: has been corrected

Alpha Pinene p-Value is unclear, 3 digits, is it lower 0.0001?

Response: Yes, is it lower 0.0001 - has been corrected

474: 2-furanmethanol is technically not a furan but an alcohol (maybe oxidized furan)

Response: Yes, we agree, but 2-furanmethanol can also be classified as a furan derivative and we have corrected this in Table 1 and in Section 4. Discussion.

500: reference after "Cotter and Hopfer (2018)" not at the end of the sentence

Response: 60.   Cotter, A.R.; Hopfer, H. The Effects of Storage Temperature on the Aroma of Whole Bean Arabica Coffee Evaluated by Coffee Consumers and HS-SPME-GC-MS. Beverages 2018, 4, 68. https://doi.org/10.3390/beverages4030068.

Reviewer 3 Report

Comments and Suggestions for Authors

Thank you for your responses.

Author Response

Dear Reviewer,

I would like to express my sincere gratitude for reviewing my article once again. Your input has significantly contributed to improving the quality of the work. I am grateful for the time and effort you have dedicated to the review process.

Best regards,

Magdalena Gantner
